# Systems-wide analysis revealed shared and unique responses to moderate and acute high temperatures in the green alga *Chlamydomonas reinhardtii*

Ningning Zhang[1,7], Erin M. Mattoon[1,2,7], Will McHargue [1,6], Benedikt Venn [3], David Zimmer[3], Kresti Pecani[4], Jooyeon Jeong[1], Cheyenne M. Anderson[1,6], Chen Chen[5], Jeffrey C. Berry[1], Ming Xia[1], Shin-Cheng Tzeng [1], Eric Becker[1], Leila Pazouki[1], Bradley Evans[1], Fred Cross[4], Jianlin Cheng[5], Kirk J. Czymmek [1], Michael Schroda [3], Timo Mühlhaus[3] & Ru Zhang [1✉]

Different intensities of high temperatures affect the growth of photosynthetic cells in nature. To elucidate the underlying mechanisms, we cultivated the unicellular green alga *Chlamydomonas reinhardtii* under highly controlled photobioreactor conditions and revealed systems-wide shared and unique responses to 24-hour moderate (35°C) and acute (40°C) high temperatures and subsequent recovery at 25°C. We identified previously overlooked unique elements in response to moderate high temperature. Heat at 35°C transiently arrested the cell cycle followed by partial synchronization, up-regulated transcripts/proteins involved in gluconeogenesis/glyoxylate-cycle for carbon uptake and promoted growth. But 40°C disrupted cell division and growth. Both high temperatures induced photoprotection, while 40°C distorted thylakoid/pyrenoid ultrastructure, affected the carbon concentrating mechanism, and decreased photosynthetic efficiency. We demonstrated increased transcript/protein correlation during both heat treatments and hypothesize reduced post-transcriptional regulation during heat may help efficiently coordinate thermotolerance mechanisms. During recovery after both heat treatments, especially 40°C, transcripts/proteins related to DNA synthesis increased while those involved in photosynthetic light reactions decreased. We propose down-regulating photosynthetic light reactions during DNA replication benefits cell cycle resumption by reducing ROS production. Our results provide potential targets to increase thermotolerance in algae and crops.

[1] Donald Danforth Plant Science Center, St. Louis, Missouri 63132, USA. [2] Plant and Microbial Biosciences Program, Division of Biology and Biomedical Sciences, Washington University in Saint Louis, St. Louis, Missouri 63130, USA. [3] TU Kaiserslautern, Kaiserslautern 67663, Germany. [4] The Rockefeller University, New York, New York 10065, USA. [5] University of Missouri-Columbia, Columbia, Missouri 65211, USA. [6] Present address: Plant and Microbial Biosciences Program, Division of Biology and Biomedical Sciences, Washington University in Saint Louis, St. Louis, Missouri 63130, USA. [7] These authors contributed equally: Ningning Zhang, Erin M. Mattoon. ✉email: rzhang@danforthcenter.org

High temperatures occur frequently in nature and impair crop yields and algal biofuel production[1,2]. Global warming increases the intensity, duration, and frequency of high temperatures above the optimal range for plant growth. It is projected that for every degree Celsius mean global temperature increases, the yields of major crop species will decrease by 3% ~ 8%[1,3]. Photosynthetic organisms experience different intensities of high temperatures in field conditions. Many crop species have threshold temperatures between 25 and 40°C, above which reduced growth is observed; most heat stress experiments have been conducted at acutely high temperatures near 42°C or above[4]. Plants frequently experience sustained moderate high temperatures around 35°C in nature, however these conditions have been largely understudied[5]. The acute high temperature at or above 40°C is more damaging but usually less frequent or shorter-lasting than the moderate high temperature in the field. We hypothesize that plants can acclimate to moderate high temperature but have reduced acclimation capacity to acute high temperature. Additionally, we propose that different levels of high temperatures induce shared and unique responses in photosynthetic cells. Understanding how photosynthetic cells respond to and recover from different intensities of high temperatures is imperative for improving crop thermotolerance[6].

High temperatures are known to have a wide variety of impacts on photosynthetic cells. Heat-increased membrane fluidity has been proposed to activate membrane-localized mechanosensitive ion channels leading to increased intracellular calcium concentrations, which may cause signaling cascades to activate heat shock transcription factors (HSFs)[7–9]. HSFs act in the nucleus to increase transcription of genes involved in heat response, e.g., heat shock proteins (HSPs)[10–12]. A recent work proposed that the accumulating cytosolic unfolded proteins, rather than changes in membrane fluidity, trigger the expression of HSPs in green algae[13]. Furthermore, high temperatures can decrease the stability of RNAs and alter the transcriptomic landscape of cells under heat stress[14]. Additionally, high temperature can cause damage to photosynthetic electron transport chains, reducing photosynthetic efficiency[15–19], and leading to increased reactive oxygen species (ROS) accumulation[4,20,21]. Heat-induced ROS production increases DNA damage and the need for DNA repair pathways, although the mechanisms of these processes are poorly understood[10,22]. In contrast to the extensive research on the effects during heat, how photosynthetic cells recover from heat is less studied[10].

Algae have great potential for biofuel production and bioproduct accumulation, but the knowledge surrounding mechanisms of algal heat responses are largely limited as compared to land plants[10]. Outdoor algal ponds frequently experience supra-optimal temperatures at or above 35°C during summer time[23], but how algal cells respond to moderate high temperatures remains largely understudied. Many previous algal heat experiments were conducted in flasks incubated in hot water baths (at or above 42°C) with sharp temperature switches, e.g., by resuspending centrifuged cells in prewarmed medium to initiate the heat treatments[13,18]. The previous research was valuable for paving the road to understand algal heat responses. Nevertheless, high temperatures in nature, especially in aquatic environments, often increase gradually and the rate of temperature increase may affect heat responses[6]. Acute high temperature at 39°C or 42°C results in algal cell cycle arrest[18,24–26]. Long-term experiments at moderate high temperatures that do not lead to a sustained cell cycle arrest cannot be conducted in flasks because cultures grow into stationary phase, causing nutrient and light limitation and therefore complicating analyses. Consequently, investigating algal heat responses under well-controlled conditions in photobioreactors (PBRs) with turbidostatic modes can mimic the

heating speed in nature and reduce compounding factors during high-temperature treatments, improving our understanding of algal heat responses.

The unicellular green alga, *Chlamydomonas reinhardtii* (Chlamydomonas throughout), is an excellent model to study heat responses in photosynthetic cells for many reasons, including its fully sequenced haploid genome, unicellular nature allowing for homogenous treatments, generally smaller gene families than land plants, and extensive genetic resources[27–32]. At the cellular level, Chlamydomonas has many similarities with land plants, making it a powerful model organism to identify novel elements with putative roles in heat tolerance with implications for crops[10].

A previous transcriptome and lipidome-level analysis in Chlamydomonas under acute high temperature (42°C) over 1 hour (h) revealed changes in lipid metabolism and increased lipid saturation as one of the early heat responses[33]. Additionally, a proteome and metabolome-level analysis of acute high temperature (42°C) for 24-h followed by 8-h recovery demonstrated temporally resolved changes in proteins, metabolites, lipids, and cytological parameters in Chlamydomonas[18]. Both publications contributed to the foundational knowledge of how Chlamydomonas responds to acute high temperature. However, a temporally resolved transcriptome analysis during and after heat over a relatively long time is lacking and the correlation between transcriptome, proteome, and physiological responses to different intensities of high temperatures remains elusive. Integrating these multiomics approaches with physiological measurements under high-temperature conditions has great potential for improving algal thermotolerance[34]. Additionally, previous research showed increased starch accumulation when Chlamydomonas cells were switched from 30°C to 39°C heat, which is linked to cell cycle arrest and the shift from energy usage for cell cycle operation to chemical energy storage at 39°C[25,26]. However, the effects of moderate high temperature on starch accumulation remain elusive.

We investigated the response of wild-type Chlamydomonas cells to moderate (35°C) or acute (40°C) high temperatures at transcriptomic, proteomic, cytological, photosynthetic, and ultrastructural levels over a 24-h heat followed by 48-h recovery period in PBRs under well-controlled conditions. Our results showed that some of the responses were shared between the two treatments and the effects of 40°C were typically more extensive than 35°C; however, 35°C induced a unique set of responses that were absent under 40°C. Both 35 and 40°C induced starch accumulation but due to distinct mechanisms. We showed that 35°C transiently inhibited cell cycle followed by synchronization while 40°C halted the cell cycle completely. Heat at 40°C but not 35°C reduced photosynthetic efficiency, increased ROS production, and altered chloroplast structures. Furthermore, with the time-resolved paired transcriptome and proteome dataset, we demonstrated increased transcript and protein correlation during high temperature which was reduced during the recovery period. Additionally, we revealed up-regulation of genes/proteins related to DNA replication and down-regulation of those related to photosynthetic light reactions during early recovery after both heat treatments, suggesting potential crosstalk between these two pathways when resuming the cell cycle. These data further our understanding of algal heat responses and provide novel insights to improve thermotolerance of algae and crops.

## Results

**Heat at 35°C increased algal growth while 40°C largely reduced it.** We cultivated Chlamydomonas cultures under well-controlled conditions (light, temperature, air flow, and

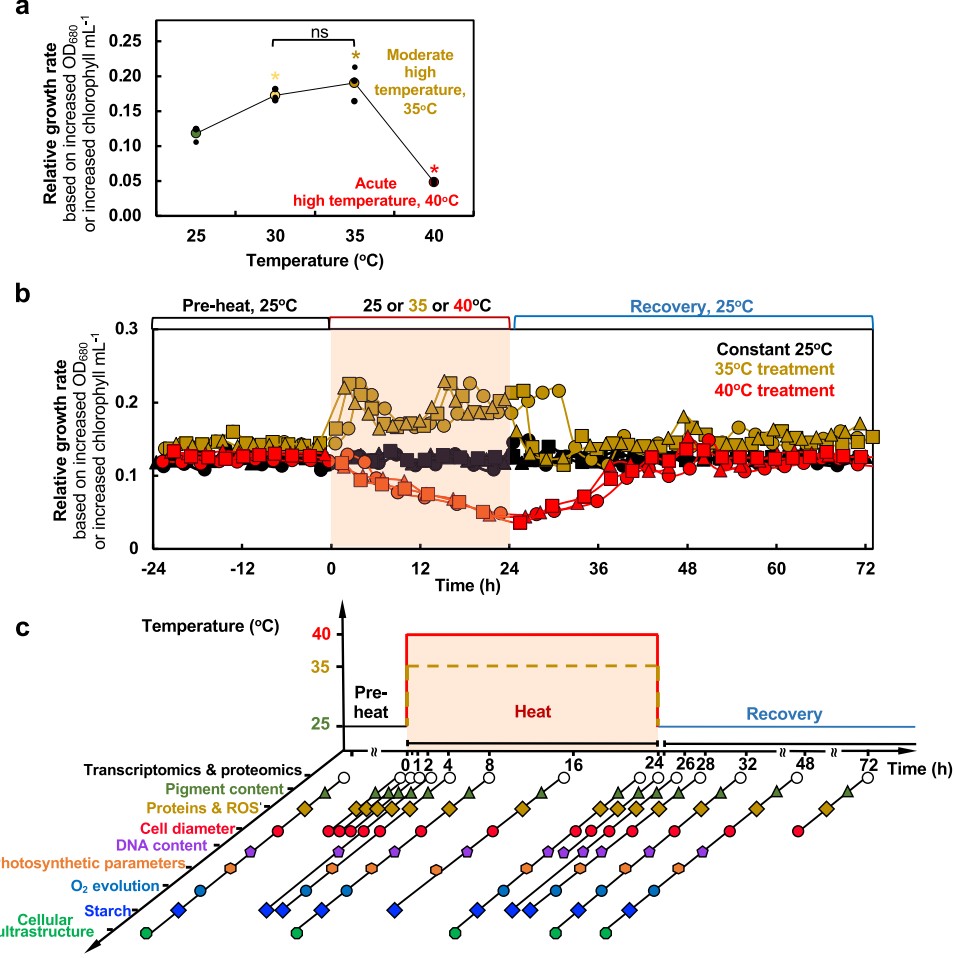

**Fig. 1 Moderate (35°C) and acute (40°C) high temperatures had contrasting effects on the growth rates of Chlamydomonas cells. a** Chlamydomonas growth rate plateaued around 35°C but was largely reduced at 40°C. Chlamydomonas cells (CC-1690, 21gr, wildtype) were grown in photobioreactors (PBRs) in Tris-acetate-phosphate (TAP) medium under turbidostatic conditions at different temperatures with a light intensity of 100 μmol photons $m^{-2} s^{-1}$ and constantly bubbling of air. Relative growth rates were calculated based on the cycling of $OD_{680}$, see Supplementary Fig. 1 and methods for details. $OD_{680}$ is proportional to total chlorophyll content in unit of μg chlorophyll $mL^{-1}$. Each temperature treatment was conducted in an individual PBR. Mean ± SE, $n = 3$ biological replicates. Statistical analyses were performed using two-tailed t-test assuming unequal variance by comparing treated samples with pre-heat or 30°C with 35°C (*, $p < 0.05$, the colors of asterisks match the treatment conditions). Not significant, ns. **b** Heat treatment at 35°C (brown) increased growth rates while 40°C (red) reduced it. Algal cultures in separate PBRs were first acclimated at 25°C for 4 days before the temperature was switched to 35°C or 40°C for 24 h, followed by recovery at 25°C for 48 h. Algal cultures grown constantly at 25°C (black) served as controls, which demonstrated steady growth without heat treatments. Three independent biological replicates for each condition were plotted. **c** PBR cultures with different treatments were sampled at a series of time points to study heat responses at multiple levels. Symbol colors match the colors of the parameters assayed. **b, c** The red shaded areas depict the duration of high temperature.

nutrient availability with supplied carbon source, acetate) by using turbidostatic mode based on $OD_{680}$ in photobioreactors (PBRs). $OD_{680}$ is proportional to total chlorophyll content in the unit of μg chlorophyll $mL^{-1}$. Fresh medium was added to the culture automatically by a peristaltic pump when the $OD_{680}$ reached the defined maximum value to dilute the culture and the pump was stopped when the $OD_{680}$ dropped to the defined minimum value (Supplementary Fig. 1a, b). Algal cultures then grew at approximately exponential rate to the defined maximum $OD_{680}$ value before the next dilution cycle. The turbidostatic mode precisely controlled the growth condition. We calculated the doubling time or relative growth rates (inverse of the doubling time) based on the exponential growth phase. The doubling time and relative growth rates we refer to throughout are based on the increase of $OD_{680}$ or the total chlorophyll per mL culture.

The relative growth rate of Chlamydomonas cells in PBRs grown mixotrophically increased with higher temperature between

25°C and 30°C, plateaued at 35°C, and largely decreased at 40°C, as compared to the control 25°C (Fig. 1a). Hence, we defined 35°C as moderate high temperature and 40°C as the acute high temperature under our experimental conditions. To investigate the systems-wide responses of Chlamydomonas to moderate and acute high temperatures, we acclimated algal cells in PBRs with well-controlled conditions at constant 25°C, followed by 24-h heat treatment at 35 or 40°C, and recovery at 25°C for 48 h (Fig. 1b, Supplementary Fig. 1a–c). Temperature increases from 25 to 35 or 40°C took about 30 minutes (min), and neither heat treatment affected cell viability (Supplementary Fig. 1d, e). In contrast, we found that a sharp temperature switch without gradual temperature increases reduced cell viability (Supplementary Fig. 1f), demonstrating heating speed affects thermotolerance. The transcript levels of selected circadian regulated genes, LHCA1[35] and TRXF2[36], did not change significantly under constant 25°C (Supplementary Fig. 1g, h), suggesting minimal circadian regulation existed under our experimental conditions with turbidostatic

control and constant light. Our observed changes during and after heat were therefore most likely attributable to heat treatments.

Based on the rate of chlorophyll production, algal growth increased during cultivation at 35°C and decreased at 40°C compared to 25°C. The increased growth at 35°C was confirmed by medium consumption rates and growth on plates (Supplementary Fig. 2). We harvested the PBR cultures throughout the time-course experiment for systems-wide analyses, including transcriptomics, proteomics, cell physiology, photosynthetic parameters, and cellular ultrastructure (Fig. 1c). RT-qPCR analysis of select time points showed that both 35°C and 40°C induced heat stress marker genes (*HSP22A* and *HSP90A*) and *HSF*s, suggesting both heat treatments could induce heat responses, although the induction amplitude was much larger under 40°C than 35°C (up to 20-fold, Supplementary Fig. 3a–d). RNA-seq data were verified by testing select genes with RT-qPCR, with highly consistent results between the two methods (Supplementary Fig. 3).

**Transcriptomic and proteomic analyses identified shared and unique responses during and after 35°C and 40°C treatments.** Two-dimensional Uniform Manifold Approximation and Projection (UMAP) of Transcripts per Million (TPM) normalized RNA-seq data resulted in three distinct clusters (Fig. 2a). The cluster during heat had a temporally resolved pattern showing increasing variance between 35°C and 40°C time points throughout the high temperature treatment. The early recovery cluster consisted of 2- and 4-h recovery samples after 35°C heat, as well as 2-, 4-, and 8-h recovery samples after 40°C heat. Late recovery and pre-heat samples clustered together, suggesting transcriptomes fully recovered by 8-h following 35°C and 24-h following 40°C treatment. UMAP of proteomics data results in two distinct clusters, separating the 35°C and 40°C treated samples and demonstrating temporally resolved proteomes (Fig. 2b). However, the samples during heat, early, and late recovery did not fall into their own distinct clusters, consistent with resistance to rapid changes on the protein level as compared with the transcript level. The proteome recovered to pre-heat levels by the 48-h recovery after both 35°C and 40°C.

We employed differential expression modeling to identify differentially expressed genes (DEGs) that were overlappingly or uniquely up- or down-regulated during and after 35°C or 40°C heat treatments (Fig. 2c, Supplementary Data 1). The greatest number of DEGs were identified at 2- and 4-h recovery time points of 40°C treatment, while investigating the distribution of log₂(fold-change) values for DEGs showed the greatest level of up-regulation at 0.5-h and down-regulation at 16-h of heat at 40°C (Supplementary Fig. 4a). Overall, there were more DEGs at most time points in the 40°C treatment as compared to the 35°C treatment (Fig. 2c). Analysis of differentially accumulated proteins (DAPs) that were overlappingly or uniquely up- or down-regulated during or after 35°C and 40°C showed a smoother distribution of expression pattern than transcriptomic data (Fig. 2d, Supplementary Data 2). Increasing numbers of DAPs were identified throughout the high-temperature period, followed by a gradual reduction throughout the recovery period. The distribution of log₂(fold-change) for DAPs at each time point also showed a smoother pattern than for transcriptome data (Supplementary Fig. 4b). Through transcriptomic and proteomic analyses, we identified shared and unique responses for the 35°C and 40°C treatment groups (Supplementary Fig. 4c–f, Supplementary Data 1 and 2).

The global transcriptome analysis revealed the three most dominant transcriptional patterns during the two treatments (Supplementary Fig. 5a, b). The first constraint (λ1) divided the

transcripts into acclimation and de-acclimation phases; the second constraint (λ2) separated control from disturbed conditions; the third constraint (λ3) showed a more fluctuating fine regulation. The amplitude difference between the treatments at 35°C and 40°C suggests an overall higher regulatory activity during the 40°C than 35°C treatment. However, there were a set of 108 genes uniquely upregulated during 35°C but not 40°C heat (Supplementary Fig. 4c, Supplementary Data 1), including *GAPDH* (involved in gluconeogenesis, glycolysis, and Calvin-Benson Cycle), *TAL1* (involved in pentose phosphate pathway), *COX15* (involved in mitochondrial assembly), and *CAV4* (encoding a putative calcium channel) (Supplementary Fig. 5c–f). Additionally, when investigating the log₂(fold-change) ratios of overlapping up- or down-regulated genes in both treatment groups at the same time point, we found that although many genes had higher differential expression with 40°C than 35°C treatment, some genes were more highly differentially expressed in the 35°C than the 40°C treatment group (Supplementary Fig. 6, Supplementary Data 1). Taken together, these results indicate that 35°C induces a unique set of responses in Chlamydomonas that have not been previously described.

Of the 3,960 heat-induced genes (HIGs, up-regulated in at least one-time point of 35°C or 40°C), 2,754 were present in the JGI InParanoid ortholog list[37,38]. We used these data to investigate the conservation of Chlamydomonas HIGs with *Volvox carteri* (Volvox), *Arabidopsis thaliana* (Arabidopsis), *Oryza sativa* (rice), *Triticum aestivum* (wheat), *Glycine max* (soybean), *Zea mays* (maize), *Sorghum bicolor* (sorghum), and *Setaria viridis* (Setaria). For most plant species tested, approximately 1,000 Chlamydomonas HIGs have orthologs (Supplementary Fig. 5g). Between Chlamydomonas and the model plant Arabidopsis, there are 509 HIGs with a one-to-one orthologous relationship (Supplementary Fig. 5g, Supplementary Data 1).

In our transcriptome and proteome data, 44.7% of the transcripts that met minimum read count cutoffs have MapMan annotations, and 80.4% of proteins have MapMan annotations (Supplementary Fig. 4g, 4h). MapMan functional enrichment analysis of DEGs at each time point of the 35°C or 40°C treatment showed that early induced shared responses to both heat treatments included canonical heat response pathways, protein folding, and lipid metabolism (Supplementary Table 1, Supplementary Data 3). MapMan terms related to DNA synthesis, cell motility, protein processing, and RNA regulation were enriched in overlapping gene sets down-regulated during most time points of both 35°Cand 40°C heat treatments. DNA synthesis and repair MapMan terms were significantly enriched in genes up-regulated in both 35°C and 40°C treated samples during the 2- and 4-h recovery time points. MapMan terms related to amino acid metabolism, mitochondrial electron transport, and purine synthesis were enriched in gene sets uniquely up-regulated during 35°C heat. Carbon fixation (e.g., carbon concentrating mechanism) and starch synthesis related MapMan terms were significantly enriched in gene sets uniquely up-regulated during 40°C heat. MapMan terms related to amino acid metabolism and mitochondrial electron transport were enriched in gene sets uniquely down-regulated during early heat of 40°C, in contrast to 35°C.

**Transcript/protein correlation increased during heat but decreased during recovery.** Investigation of Pearson correlation coefficients between log₂(fold-change) values for transcripts and proteins grouped by MapMan functional categories revealed higher positive correlation between transcriptome and proteome during heat than recovery for both 35°C and 40°C treatments (Fig. 2e, f, Supplementary Fig. 7). This indicates that

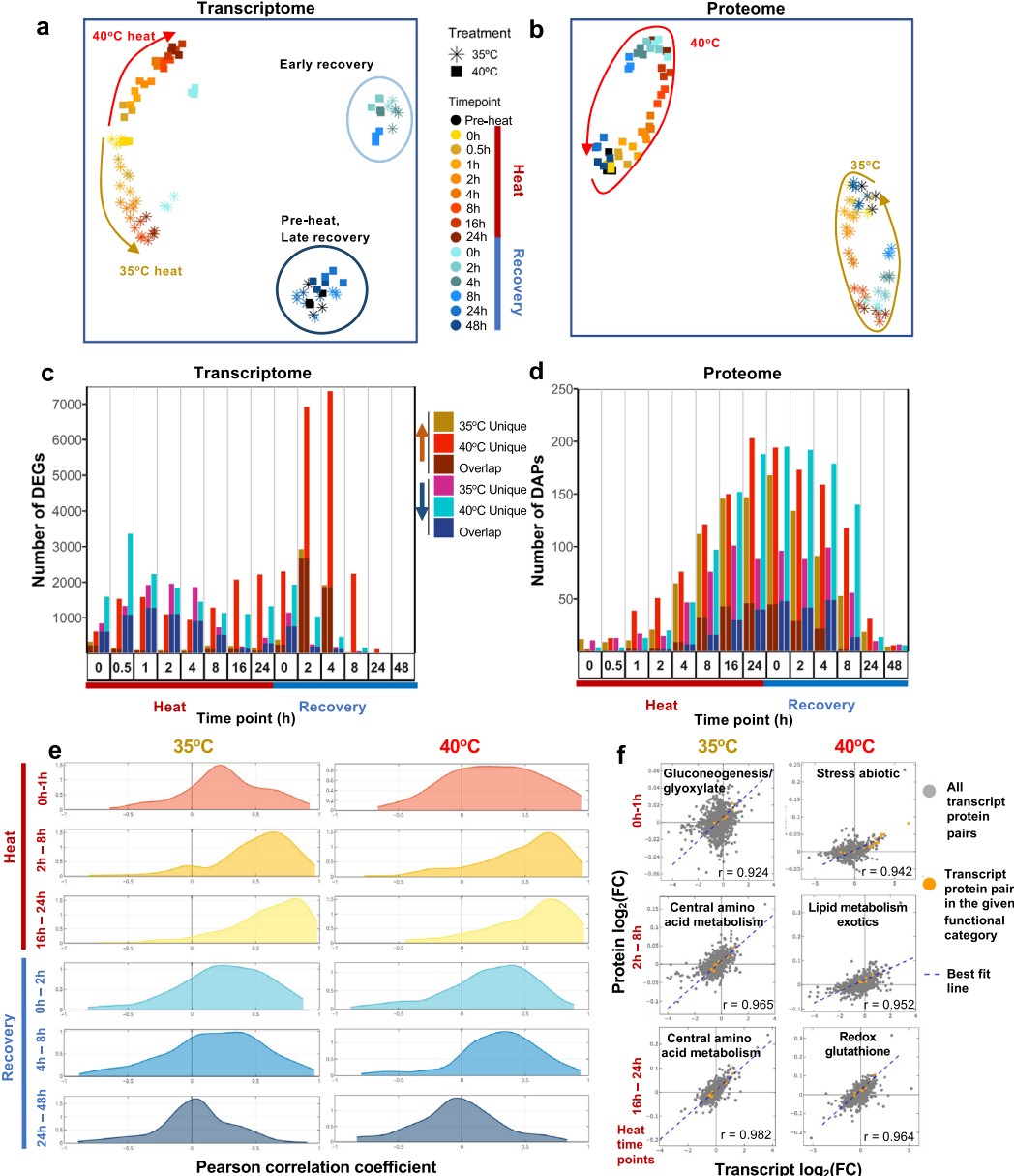

**Fig. 2 Transcriptomics and proteomics revealed distinct and dynamic responses during and after heat treatments of 35°C and 40°C. a** Uniform Manifold Approximation and Projection (UMAP) of Transcripts Per Million (TPM) normalized RNA-seq read counts and **(b)** UMAP of normalized protein intensities. Each data point represents all normalized counts from a single sample. Stars and squares represent algal samples with heat treatments of 35°C or 40°C, respectively. Different colors represent different time points. Brown and red arrows show the movement through time of the 35°C and 40°C treated samples, respectively. **c**, **d** Number of unique or overlapping Differentially Expressed Genes (DEGs) and Differentially Accumulated Proteins (DAPs) with heat treatment of 35°C or 40°C at different time points, respectively. Each time point has four bars: the first bar represents genes up-regulated in 35°C, the second represents genes up-regulated in 40°C, the third represents genes down-regulated in 35°C, and the fourth represents genes down-regulated in 40°C. The bottom portion of the stacked bars represents genes/proteins that are differentially expressed in both treatment groups and the top portion represents genes that are uniquely differentially expressed in the given treatment group at that time point. Significant differential expression in transcriptome data was defined as absolute values of $\log_2$(fold-change, FC) > 1, false discovery rate (FDR) < 0.05, and absolute difference of TPM normalized read counts between treatment and pre-heat control ≥ 1. Significant differential accumulation of proteins was defined by Dunnett's FWER < 0.05. **e** Analysis of correlation between transcripts and proteins revealed overall rising positive correlation during the heat treatment and a decreasing correlation during recovery. The time points were subdivided into three heat (red labels) and three recovery (blue labels) windows for 35°C (left) and 40°C (right) treated samples. The density plots of Pearson correlation coefficients between the fold-changes of transcripts and proteins are shown. X, Pearson correlation coefficient. Y, frequency density. Time points during heat: 0 h, reach high temperature of 35°C or 40°C; 1 h, heat at 35°C or 40°C for 1 h, similar names for other time points during heat. Time points during recovery: 0 h, reach control temperature of 25°C for recovery after heat; 2 h, recovery at 25°C for 2 h, similar names for other time points during recovery. See Supplementary Fig. 7 for more information. **f** Scatter plots of all transcript-protein pairs at 35°C (left) and 40°C (right) for the three heat time point bins shown in (**e**). X and Y, transcript and protein $\log_2$FC, compared with pre-heat, respectively. Transcript-protein pairs are shown as gray dots. Best fit lines are shown in blue. The Pearson correlation coefficient is shown at the bottom right corner of each scatterplot. Transcript-protein pairs belonging to MapMan functional categories with the highest Pearson correlation coefficient in the given time point bin are shown in orange. See the interactive figures in Supplementary Data 9: transcripts/proteins correlation.

transcriptional regulation dominates the heat period. Further investigation of Pearson correlation coefficients for individual MapMan terms showed that functional categories had varying correlation values throughout the course of high temperatures and recovery (Supplementary Data 4, 9). In early time points of the 35°C treatment, the gluconeogenesis/glyoxylate-cycle functional category had the highest correlation between transcripts and proteins, followed by amino acid and lipid metabolism at later heat time points (Fig. 2f). In early time points of the 40°C treatment, the abiotic stress functional category had the highest correlation between transcripts and proteins, followed by lipid metabolism and redox pathways at later heat time points (Fig. 2f). The proteins in the MapMan bin gluconeogenesis/glyoxylate cycle increased during 35°C but decreased during 40°C, suggesting elevated and reduced gluconeogenesis/glyoxylate-cycle activity during 35°C and 40°C heat treatment, respectively (Supplementary Fig. 8a–d). Isocitrate lyase (ICL1) is a key enzyme of the glyoxylate cycle in Chlamydomonas[39]. The transcript, protein, as well as the correlation of ICL1 increased during 35°C heat but decreased during 40°C heat (Supplementary Data 9,10, gluconeogenesis _ glyoxylate cycle.html). In Chlamydomonas, acetate uptake feeds into the glyoxylate cycle and gluconeogenesis for starch biosynthesis[40,41]. Several proteins related to acetate uptake/assimilation[41,42] were increased during 35°C heat but deceased during 40°C heat (Supplementary Fig. 8e–h, Supplementary Data 2). Our results suggested 35°C treatment increased acetate uptake/assimilation and glyoxylate cycle and gluconeogenesis pathways, which may be suppressed by the 40°C treatment.

**Network modeling of transcriptome and proteome revealed expression patterns of key pathways during and after heat treatments.** To investigate common transcriptional expression patterns throughout heat and recovery, we performed Weighted Correlation Network Analysis (WGCNA) on TPM-normalized RNA-seq data from 35°C and 40°C transcriptomes (Fig. 3a, Supplementary Data 5). Most genes with known roles in heat response belong to Transcriptomic Module 1 (TM1) with peak expression at 0.5-h heat, including *HSF*s and many *HSP*s. TM1 is also significantly enriched for MapMan terms related to protein folding, lipid metabolism, and starch synthesis. In total, there are 628 genes associated with this module, about 62% of which lack Chlamydomonas descriptions and 25% have no ontology annotations, indicative of novel genes with putative roles in heat response that have been previously undescribed. TM2 has peak expression at 1-h heat and is significantly enriched for MapMan terms relating to the TCA cycle and carbon concentrating mechanism. TM3 has peak expression at the beginning of both heat and recovery periods and is significantly enriched for MapMan terms related to RNA regulation and chromatin remodeling, highlighting the extensive alterations in transcriptional regulation caused by changing temperatures. TM4 contains 137 genes that have sustained increased expression throughout 35°C, with slightly reduced or no change in expression in 40°C. This module is significantly enriched for MapMan terms related to amino acid metabolism, ferredoxin, and fatty acid synthesis/elongation. TM4 represents genes that are unique to 35°C responses. TM5 shows increased expression throughout the heat period at 40°C. This module is enriched for ABA synthesis/degradation, suggesting potential interaction between the ABA pathway and algal heat responses. Previous reports showed that heat treated Arabidopsis and rice leaves had increased ABA contents and exogenous ABA application improved heat tolerance in WT rice plants[43,44]. ABA is reported to be involved in tolerance to oxidative stress[45], HCO3−

uptake[46], and stress acclimation in Chlamydomonas[47]. TM6 displays similar expression patterns in both 35°C and 40°C treatments, with increased expression during early heat and reduced expression during early recovery. This module is enriched for genes in the Calvin-Benson cycle, PSII biogenesis, and tetrapyrrole synthesis. TM7 shows increased expression in 40°C uniquely in late heat, which continues through the early and mid-recovery periods. This module is notably enriched for protein degradation, ubiquitin E3 RING, and autophagy, which may indicate the need to degrade certain proteins and cellular components during and after prolonged acute high temperature at 40°C. TM8 shows increased expression only at the 0-h recovery time point immediately after the cooling from the 35°C and 40°C heat to 25°C. This module is notably enriched for bZIP transcription factors, oxidases, calcium signaling, and protein posttranslational modification, which likely contribute to the broad changes observed when recovering from high temperatures. TM9 and TM10 both have peaks in expression in recovery (2 h, 4 h for TM9; and 8 h for TM10) and have significantly enriched MapMan terms related to DNA synthesis. TM11 peaks in expression at 8-h recovery and is significantly enriched for MapMan terms related to protein posttranslational modification, consistent with the decreased correlation between transcript and protein levels at late recovery stages (Fig. 2e). TM12 shows reduced expression during the early recovery period and is significantly enriched for MapMan terms related to photosynthetic light reactions and protein degradation.

We verified the expression patterns of several key pathways of interest and transcription factors by visualizing $\log_2$(fold-change) values from differential expression modeling (Supplementary Fig. 9, 10, Supplementary Data 6). The RNA-seq results of select pathways from differential expression modeling were highly consistent with WGCNA modeling and provided gene-level resolution of interesting trends within these pathways. The down-regulation of select transcripts involved in photosynthetic light reactions during early recovery was also verified by RT-qPCR (Supplementary Fig. 9i, j).

Network modeling was performed separately for the proteomes of the 35°C and 40°C treated samples due to differences in peptides identified through LC-MS/MS between the two treatment groups and the relatively smaller number of proteins identified compared to transcripts. This analysis identified common expression patterns (Proteomics Module, PM) in the proteome data (Fig. 3b, Supplementary Data 7). Prominent functional terms that were enriched in the respective module are given for proteins correlating positively (PM_a) or negatively (PM_b) to their corresponding eigenvector (FDR < 0.05). Protein modules that increased during 35 and 40°C heat treatments are enriched for MapMan terms related to the part of photosynthetic light reactions, protein folding, and redox (PM1_a, PM6_a, PM7_a). Proteins that increased during the recovery phase after either heat treatments are enriched for MapMan terms related to protein synthesis (PM1_b, PM7_b), DNA synthesis and chromatin structure (PM2_b, PM6_b). Proteins related to photosynthetic light reactions first decreased and then increased during the recovery of both treatments (PM1_a, PM4_b, PM8_b). Unique responses for the 35°C treatment include proteins related to mitochondrial electron transport and lipid metabolism increased during heat (PM5_a) and proteins related to RNA processing and cell organization increased during the recovery phase after 35°C heat (PM1_b, PM2_b). Unique responses for the 40°C treatment include proteins related to abiotic stress increased during heat (PM10_b) and proteins related to cell motility and RNA increased during the recovery phase after 40°C heat (PM6_b, PM7_b). Network modeling of transcriptome and proteome data yielded consistent patterns for several key pathways during and

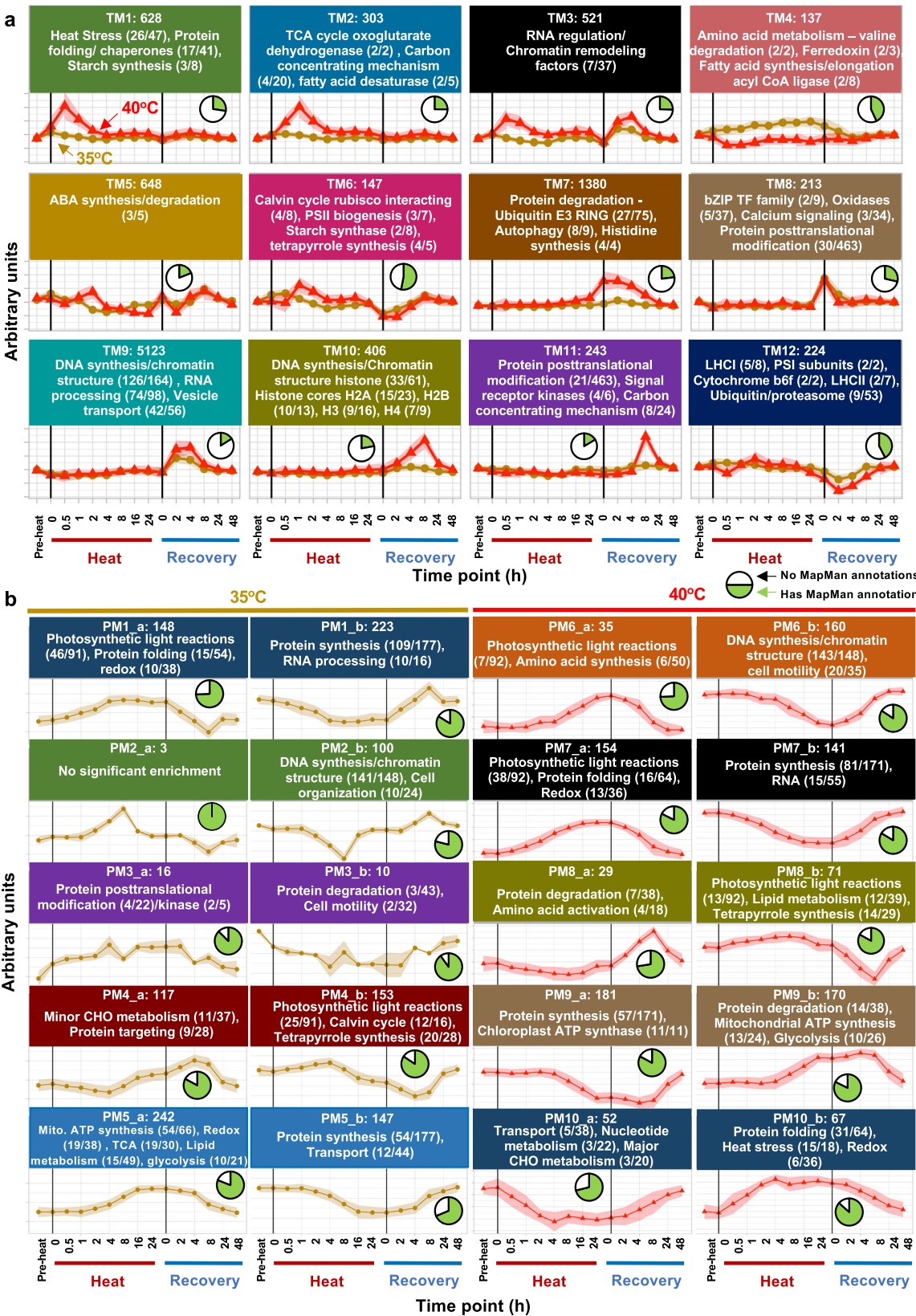

after heat treatments, e.g., heat responses, photosynthetic light reactions, and DNA synthesis.

**Heat at 35°C synchronized the cell cycle while 40°C arrested it.** The increased transcript and protein levels related to DNA

synthesis during recovery (Fig. 3) prompted us to investigate the expression pattern of cell cycle related genes because DNA synthesis takes place immediately before and during cell division in Chlamydomonas[48]. Both 35°C and 40°C disrupted the expression pattern of cell cycle genes as compared to the pre-heat level, which recovered by 8-h heat treatment at 35°C but not

**Fig. 3 Transcriptomic and proteomic network modeling revealed differential regulation of key biological pathways with treatments of 35°C or 40°C. a** Weighted correlation network analysis (WGCNA) of transcriptome data identified gene modules with similar expression patterns. TM: transcriptomic module. The z-normalized consensus gene expression patterns for 40°C (red triangle) and 35°C (brown circle) for each module are displayed. **b** Correlation networks of protein abundance over time courses for 35°C and 40°C heat treatments. PM, proteomic module. For each module, the eigenvectors as aggregated signal shape is depicted. Prominent functional terms that were enriched in the respective module are given for proteins correlating positively (PM_a) or negatively (PM_b) to their corresponding eigenvector (FDR < 0.05). **a, b** Black vertical lines indicate the start and end of the heat treatment at either 35°C or 40°C. Pre-heat, before heat treatments. Time points during heat: 0 h, reach high temperature of 35°C or 40°C; 0.5 h, heat at 35°C or 40°C for 0.5 h, similar names for other time points during heat. Time points during recovery: 0 h, reach control temperature of 25°C for recovery after heat; 2 h, recovery at 25°C for 2 h, similar names for other time points during recovery. The y axes are in arbitrary units. Background shading indicates consensus expression pattern ± sd of eigenmodule members. The number at the top of each facet (e.g., TM1: 628, PM1_a:148) represents the total number of genes/proteins significantly associated with the given module (ANOVA, FDR < 0.05, genes/proteins can only belong to a single module). Select statistically significantly enriched MapMan functional terms are displayed in each facet. Ratios after each MapMan term (e.g., heat stress, 26/47) represent the number of genes/proteins with the assigned MapMan term in the given module relative to the number of genes/proteins with the assigned MapMan term in our entire dataset. Pie charts show the fraction of genes/proteins associated with the given module that have at least one assigned MapMan functional term (green) relative to those associated with the given module but without MapMan functional terms (white). Full functional enrichment analysis can be found in Supplementary Data 5 and 7.

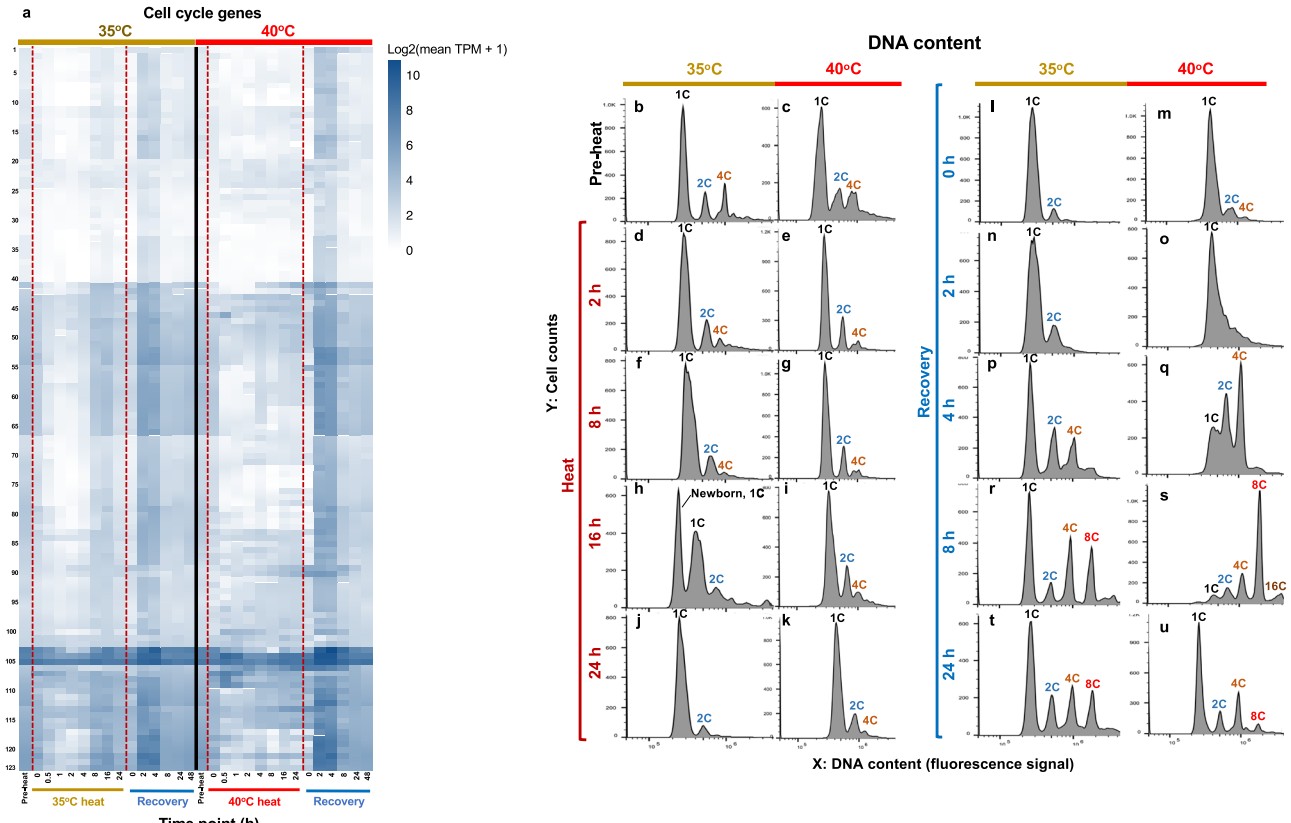

**Fig. 4 Heat at 35°C synchronized the cell cycle while heat at 40°C arrested it. a** Expression of cell cycle genes during and after heat treatment of 35°C or 40°C. Genes used for the expression pattern analysis are listed in Supplementary Data 6. Gene expression patterns for treatments of 35°C (left) and 40°C (right) are displayed as log2(mean TPM value + 1). TPM, transcripts per million. Darker blue colors indicate higher expression. Red dashed lines indicate the start and end of the heat treatments. Pre-heat, before heat treatment. Time points during heat: 0 h, reach high temperature of 35°C or 40°C; 2 h, heat at 35°C or 40°C for 2 h, similar names for other time points during heat. Time points during recovery: 0 h, reach control temperature of 25°C for recovery after heat; 2 h, recovery at 25°C for 2 h, similar names for other time points during recovery. **b–u** FACS (fluorescence-activated cell sorting) analysis of the DNA content in algal samples harvested at different time points before, during, and after heat treatment at 35°C or 40°C. For each figure panel, X axis is DNA content determined by plotting a histogram of fluorescence level (area of the fluorescence channel signal) in log-scale; Y axis is the cell counts in linear scale. DNA copy No. are labeled on top of each corresponding DNA content peak, 1 C (single DNA copy number), 2 C, 4 C, 8 C, 16 C.

during the entire 40°C heat treatment (Fig. 4a). Most cell cycle related genes had increased expression during 2- and 4-h of recovery in both treatment groups.

During the 40°C heat treatment, we observed irregular expression patterns of cell cycle related genes as compared to

pre-heat, which led us hypothesize that the cell cycle had been arrested during 40°C. To investigate this hypothesis, we quantified cellular DNA content using flow cytometry (Fig. 4b±u). Pre-heat cultures showed typical asynchronous populations: most cells had 1 C (single DNA copy number) while a small fraction of

cells had 2 C and 4 C (Fig. 4b, c). After 16-h at 35°C heat treatment, the broad 1 C size distribution from 8-h split; some of the bigger cells went to 2 C and the rest of the small cells stayed at 1 C (Fig. 4f, h). The 2 C population then divided by 24-h at 35°C, resulting in almost exclusively small 1 C cells with the same cell size as 1 C cells at pre-heat, suggesting culture synchrony (Fig. 4j). The cell division during 8-16 h of heat at 35°C was consistent with the recovery of cell cycle genes at the same time points (Fig. 4a). After 2-h of recovery at 25°C following 35°C heat, there were much fewer 2 C and 4 C cells than pre-heat, suggesting a partially synchronized population, until an almost complete recovery by 4-h at 25°C (Fig. 4l, n, p, r, t). These results indicated that the 35°C treatment synchronized cell division in Chlamydomonas.

Heat treatment at 40°C inhibited DNA replication and cell division, with three peaks of 1 C, 2 C, and 4 C persisting during all 40°C heat time points (Fig. 4c, e, g, i, k). By the end of the 24-h heat treatment at 40°C, the three cell populations had a much larger cell size, as evidenced by the right shift of the DNA peaks due to an increased cell size background effect (Fig. 4k, more information of the background effect can be found in the Methods). Cells started to replicate DNA between 2- and 4-h of recovery following 40°C heat, resulting in reduced 1 C but increased 2 C and 4 C cell populations (Fig. 4q). DNA replication continued until 8-h of recovery, resulting in the accumulation of high-ploidy level cells, ranging from 1 C to 16 C (Fig. 4s). By 24-h recovery, the cellular DNA content had almost recovered to the pre-heat level (Fig. 4u).

**Cytological parameters confirmed cell cycle arrest during 40°C heat**. Brightfield images and cell size quantification of algal cells showed that the 40°C treated cells had continuously increased cell size throughout the high-temperature period, followed by gradual recovery to the pre-stress level after returning to 25°C for 24 h (Fig. 5a, b, c). The quantity of chlorophyll, carotenoids, and protein per cell all increased during the 40°C heat treatment and gradually recovered after heat (Fig. 5d–g). ROS level per cell had a clear increasing tread in 40°C-treated cells (especially at the end of the heat and early recovery), although the changes were not significant after stringent statistical analysis with FDR correction (Fig. 5g). When normalized to cell volume, chlorophyll and carotenoid contents increased during 40°C heat treatment while no change of protein and ROS contents was observed (Supplementary Fig. 11a–d). The ratios of chlorophyll a/b and chlorophyll/carotenoid decreased during 40°C heat (Supplementary Fig. 11e, f). The changes of chlorophyll a and b were consistent with that of total chlorophyll (Supplementary Fig. 10g–j, Fig. 5d). Cell diameter, cell volume, chlorophyll and carotenoid contents only changed transiently after shifting to 35°C and after shifting back to 25°C (Fig. 5b–e, Supplementary Fig. 11).

**Heat at 40°C impaired photosynthesis while the effects at 35°C were minor**. We hypothesized that heat treatments might affect photosynthetic activities based on the changes of pigment contents (Fig. 5d, e), differentially regulated genes related to photosynthesis (Supplementary Fig. 9a, b), and the kinetics of transcripts and proteins related to the MapMan bin photosynthetic light reactions (Fig. 6, Supplementary Data 10). Most transcripts related to photosynthetic light reactions decreased during the early recovery from both heat treatments, followed by a gradual returning to the pre-stress levels (Fig. 6a, b, e, f, i, j), consistent with network modeling data (Fig. 3a). Proteins related to PSI, LHCII (light harvesting complex II), and LHCI (light harvesting complex I) increased during both heat treatments and decreased back to the pre-heat levels during the recovery (Fig. 6c, d, g, h). Proteins related to the

ATP synthase decreased during the 40°C heat treatment and the early/middle recovery of both 35°C and 40°C heat treatments (Fig. 6k, l). Overall, the kinetics of transcripts and proteins related to the MapMan bin photosynthetic light reactions showed similar trends in both 35°C and 40°C treatments.

To investigate whether these pronounced changes of proteins related to LHCI, LHCII, PSI and ATP synthase under 35°C and 40°C affected photosynthesis, we measured various photosynthetic parameters during and after the heat treatments (Figs. 7–8). The PSII efficiency and linear electron flow rates decreased during 40°C heat (especially under light intensities exceeding the growth light of 100 μmol photons $m^{-2}$ $s^{-1}$) while the 35°C heat treatment did not extensively affect these photosynthetic parameters (Fig. 7a–d). The $Q_A$ redox state reflects the balance between excitation energy at PSII and the rate of the Calvin-Benson Cycle[49,50]. The amount of reduced $Q_A$ is proportional to the fraction of PSII centers that are closed[51]. Under 35°C and 40°C, $Q_A$ had no significant changes, although 40°C-treated cells showed the trend of increased redox status (Fig. 7e, f). Both 35°C and 40°C increased the formation of NPQ; however, the increased NPQ was steady during 4-24 h of heat at 35°C while during 40°C heat, NPQ first increased to a maximum at 8-h heat, then decreased by 24-h heat (Fig. 7g, h), suggesting that accumulative heat damages under prolonged exposure to 40°C eventually exceeded the photoprotective capacity of NPQ. Relative PSII antenna size increased during the 40°C heat treatment, while it increased transiently during the 35°C heat treatment (Fig. 7i, Supplementary Fig. 12).

Additionally, we performed electrochromic shift (ECS) measurements to monitor the effects of heat on the transthylakoid proton motive force (*pmf*, estimated by $ECS_t$) and proton conductivity (Fig. 8a, b, Supplementary Fig. 13a, b). No significant changes in *pmf* and proton conductivity were observed during and after the 35°C treatment (Fig. 8a, Supplementary Fig. 13a). The *pmf* increased particularly at late time points during the 40°C treatment, followed by a slow and partial recovery after shifting cells back to 25°C. Proton conductivity decreased during and after 40°C heat treatments, suggesting reduced or compromised ATP synthase activity (Supplementary Fig. 13b), consistent with reduced abundance of proteins related to ATP synthase (Fig. 6l). During both heat treatments, NPQ formation became more sensitive to *pmf*, with higher NPQ formed at a given *pmf* compared to the pre-heat condition (Fig. 8c, d), consistent with a previous report in tobacco plants[52]. The increased sensitivity of NPQ was collapsed by the end of the 24-h heat treatment at 40°C (Fig. 8d). P700 measurement revealed that the activity of cyclic electron flow around PSI (CEF) increased during both 35°C and 40°C heat, which recovered quickly after 35°C treatment but much more slowly after 40°C treatment (Fig. 8e). P700 appeared to be more reduced during 35°C and 40°C heat, although the changes were not significant with stringent FDR correction (Supplementary Fig. 13c). Furthermore, gross photosynthetic $O_2$ evolution rates and dark respiration rates had little changes during the 35°C treatment but dropped significantly during the 40°C heat treatment (Fig. 8f, g, h). Photosynthetic parameters had no significant changes in cultures maintained under constant 25°C (Supplementary Fig. 14).

**Heat at 40°C altered thylakoid and pyrenoid ultrastructure**. The effects of high temperatures on photosynthesis prompted us to investigate cellular ultrastructure using transmission electron microscopy (TEM) (Fig. 9a–r). Thylakoids became disorganized and loosely packed in cells treated with 40°C (Fig. 9f, g). Investigation of pyrenoid ultrastructure (Fig. 9j–r, s) showed that cells

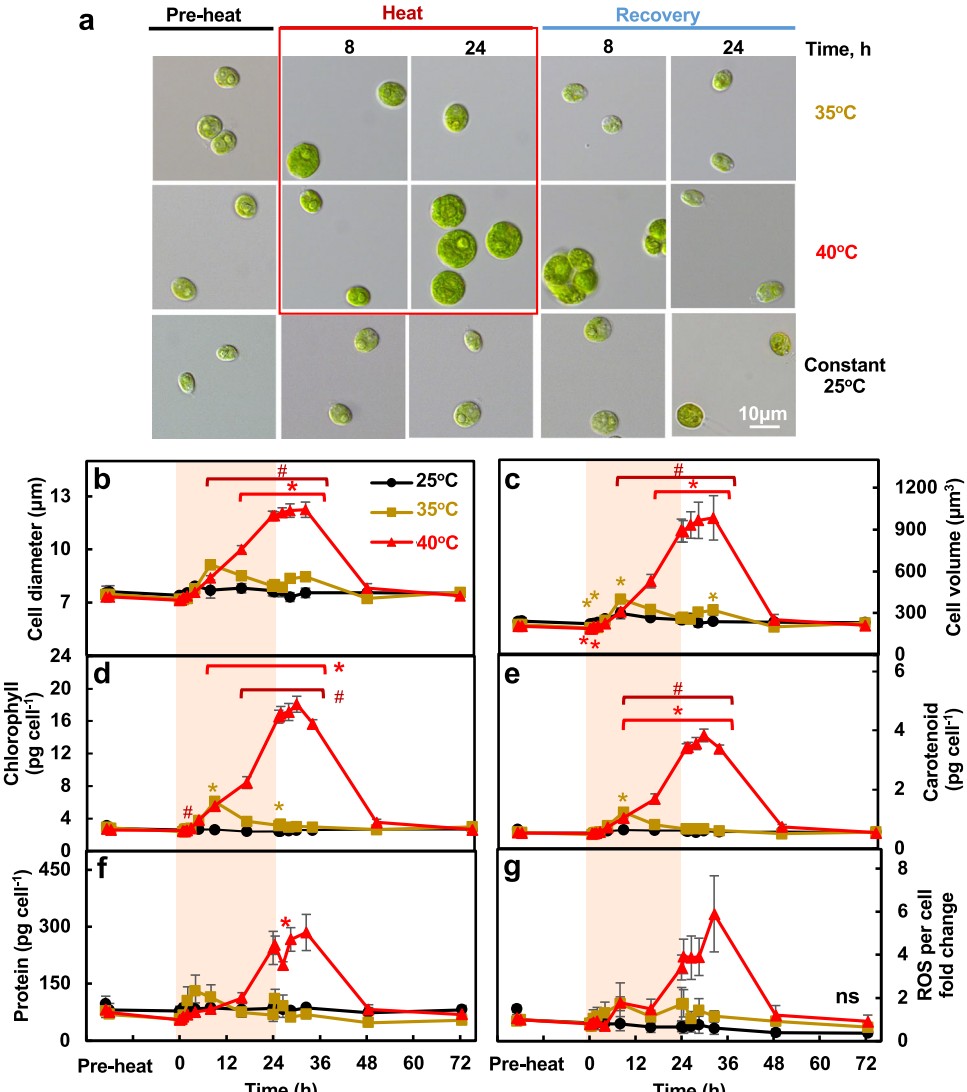

**Fig. 5 Heat of 40°C persistently increased cell size, cellular levels of pigments, and proteins while these effects were transient with 35°C heat. a** Light microscopic images of Chlamydomonas cells. **b, c** Cell diameters and volume determined using a Coulter Counter. **d–f** Total chlorophyll, carotenoid, protein content per cell. **g** Fold-change of reactive oxygen species (ROS) levels per cell quantified using CM-H2DCFDA ROS indicator. Mean ± SE, $n = 3$ biological replicates. Black, brown and red curves represent experiments with constant 25°C, treatments of 35°C or 40°C respectively. Red shaded areas depict the duration of high temperature. Statistical analyses were performed using two-tailed t-test assuming unequal variance by comparing treated samples with 25°C at the same time point (*, $p < 0.05$) or between 35°C and 40°C at the same time point (#, $p < 0.05$). **b–g** P values were corrected by FDR. The colors and positions of asterisks (*) match the treatment conditions and time points, respectively. The positions of pound signs (#) match the time points. (g) Not significant, ns, $p > 0.05$ after FDR correction.

treated with 40°C had altered pyrenoid matrices and absence of thylakoid tubules inside the pyrenoid (Fig. 9o, p), suggesting an inefficient carbon concentrating mechanisms (CCM). No changes in pyrenoid ultrastructure were observed in cells treated with 35°C (Fig. 9k, l). ImageJ quantification of pyrenoid structures showed that cells treated with 40°C had increased pyrenoid areas, which was attributed to increased areas of both the pyrenoid matrix and starch sheath (Fig. 9t, u, v). The increased pyrenoid size was abolished after 8-h of recovery. Biochemical quantification of starch contents showed that both 35°C and 40°C treatments increased starch levels per cell and per cell volume, which decreased during recovery (Fig. 9w, x). At the end of the heat treatments and early recovery, cells exposed to 40°C had a higher starch content per cell than those exposed to 35°C, but the differences between the two treatments were not significant per cell volume.

## Discussion

We investigated how Chlamydomonas cells respond to moderate (35°C) and acute (40°C) high temperature at systems-wide levels (Fig. 1c). Our results show that 35 and 40°C triggered shared and unique heat responses in Chlamydomonas (Fig. 10).

Both high temperatures induced the expression of *HSF1* and *HSF2*, as well as canonical high-temperature response genes *HSP22A* and *HSP90A*, increased cell size, chlorophyll and carotenoid contents, PSII/PSI ratio, NPQ, CEF, PSI redox state, and starch formation (Fig. 10). The changes under 35°C were often transient and moderate while those under 40°C were sustained and dramatic. The correlation between transcripts and proteins increased during both heat treatments, suggesting that responses during heat were largely transcriptionally regulated. Functional categories of gluconeogenesis/glyoxylate-cycle and abiotic stress had the highest correlation between transcripts and proteins in

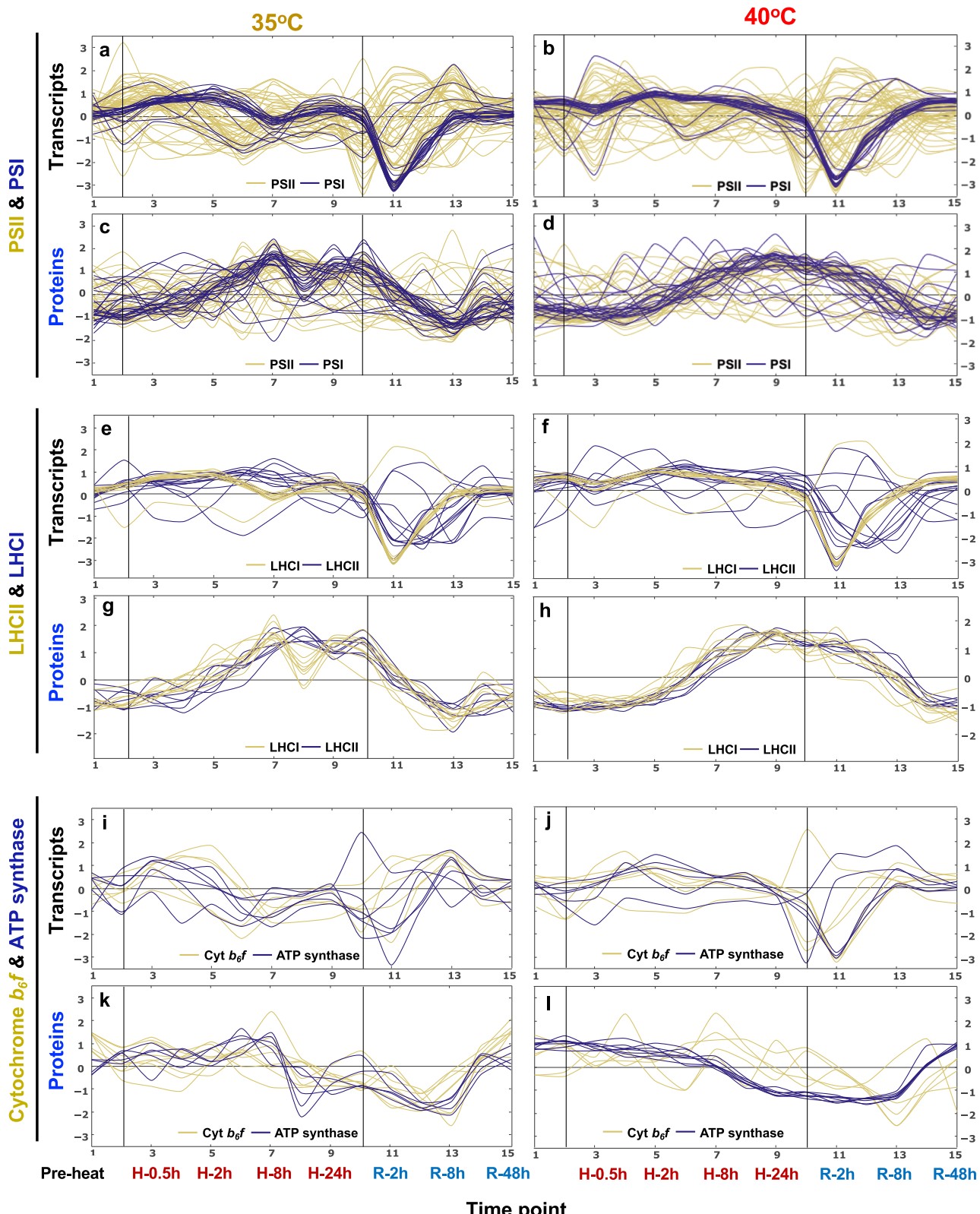

**Fig. 6 Transcripts and proteins related to photosynthetic light reactions changed dynamically during and after 35°C and 40°C heat treatments.** Signals of transcripts (**a**, **b**, **e**, **f**, **i**, **j**) and proteins (**c**, **d**, **g**, **h**, **k**, **l**) related to the MapMan bin photosynthetic light reactions, including PSII and PSI (**a–d**), LHCII and LHCI (**e–h**), Cytochrome $b_6f$ and ATP synthase (**i–l**), were standardized to z scores (standardized to zero mean and unit variance) and plotted against equally spaced time point increments. The black vertical lines indicate the start and end of heat treatments at 35°C (**a**, **c**, **e**, **g**, **i**, **k**) and 40°C (**b**, **d**, **f**, **h**, **j**, **l**), respectively. Time points are labeled at the bottom. Timepoint 1: pre-heat. Time points 2-9, heat treatment at 35°C or 40°C, including reaching high temperature (0), 0.5, 1, 2, 4, 8, 16, 24 h during heat; time points 10–15, recovery phase after heat treatment, including reaching control temperature (0), 2, 4, 8, 24, 48 h during recovery. See the interactive figures with gene IDs and annotations in Supplementary Data 10 (transcript/protein dynamics), the groups of PS.lightreaction (PS for photosynthesis).

early time points of 35°C and 40°C treatment (Fig. 2f), respectively. High correlation values of these functional categories indicate that these responses may occur rapidly without much post-transcriptional regulation, which may help coordinate activities to adapt to high temperature quickly and efficiently. The decreased correlation between transcripts and proteins during both recoveries was consistent with increased protein posttranslational modification after heat treatments based on the RNA-seq network modeling results (Fig. 3a, TM8/11).

The increased NPQ in cells treated with 35°C and 40°C heat suggested that both heat treatments compromised photosynthetic efficiency (Fig. 7g, h). Heat at 40°C reduced photosynthetic efficiency much more than 35°C, especially when photosynthetic parameters were evaluated with light intensities higher than the growth light of 100 μmol photons $m^{-2}$ $s^{-1}$ (Figs. 7 and 8). Under the growth light we employed, the differences in photosynthetic parameters between 35°C and 40°C were smaller with comparable values, consistent with the similar kinetics of transcripts and proteins related to the MapMan bin photosynthetic light reactions (Fig. 6). We conclude that in algal cultures grown in PBRs under a growth light of 100 μmol photons $m^{-2}$ $s^{-1}$, both 35°C and 40°C heat treatments affected photosynthetic efficiency but photosynthetic activity was maintained at a comparable level during both heat treatments; however, increasing light intensities exaggerated the heat-induced damages to photosynthesis, especially with the 40°C treatment.

During early recovery from both heat treatments, transcripts and proteins related to DNA synthesis increased while those related to photosynthetic light reactions decreased (Fig. 3). In synchronized algal cultures under day/night cycles, genes related to DNA synthesis and cell cycle peak during the early dark phase when the genes related to photosynthetic light reactions had minimal expression[53,54]. However, under constant light as in our experiment, genes related to DNA synthesis and photosynthetic light reactions may express simultaneously but their quantitative expression manner under constant light is understudied. The induction of cell cycle genes after recovery were comparable after both 35°C and 40°C heat (Fig. 4a), but the significant down-regulation of genes related to photosynthetic light reactions only occurred in the recovery following 40°C (Supplementary Fig. 9a). The photosynthetic light reactions are a major source of ROS production[55]. The measured ROS levels reflect the competition between ROS production and scavenging. Although the increase of the measured ROS level was not significant, the up-regulation of ROS scavenging transcripts (Supplementary Fig. 9e) during the heat and early recovery phase of 40°C supported the increased ROS production with 40°C treatment. Synchronized cells at the dark phase of day/night cycle most likely have minimal ROS production, as evidenced by the down-regulation of many ROS response genes[53]. Cells with 40°C heat had very different physiologies from synchronized culture at the early dark phase under the control temperature, considering the increased ROS production and heat damaged cellular structures in 40°C-treated cells. Thus, we suspect the up-regulation of transcripts related to cell cycle and down-regulation of transcripts related to photosynthetic light reactions during the recovery from 40°C under constant light and during the dark phase of day/night-cycle may be due to different mechanisms. ROS accumulation impairs DNA replication and induces DNA damage[56,57]. Mammalian cells are more sensitive to high temperatures during DNA replication than other cell cycle stages[58]. We propose that down-regulation of photosynthetic light reactions during DNA synthesis is beneficial to resuming the cell cycle by reducing ROS production during the early recovery. The mechanisms of the opposite transcriptional regulation of DNA replication and photosynthetic light reactions during the recovery from high temperatures is unknown but

interesting for future research. One form of ROS is $H_2O_2$, which is highly diffusible and stable[55]. Recently, Niemeyer et al. employed hypersensitive $H_2O_2$ sensors in different compartments of Chlamydomonas cells and showed that $H_2O_2$ levels increased in the nucleus after heat treatment, suggesting diffusion of $H_2O_2$ from other cellular compartments to the nucleus[21]. $H_2O_2$ has been proposed as a secondary messenger in signal transduction[59]. Increased $H_2O_2$ in the nucleus may affect gene expression, e.g., those involved in photosynthetic light reactions.

Both high temperatures affected cell cycle genes, but during 35°C cells could recover the expression of cell cycle genes to the pre-heat level after 8-h of heat treatment with a partially synchronized cell cycle at the end of the 24-h heat treatment (Fig. 4). In contrast, under the 40°C heat treatment, the expression of cell cycle genes was disrupted. Additionally, we observed down-regulation of genes encoding cell wall proteins during both 35°C and 40°C heat treatments and many of these genes were up-regulated during the recovery of 35°C and 40°C; the differential regulation of cell wall genes was more dramatic with 40°C than 35°C treatment (Supplementary Fig. 9h). The Chlamydomonas cell wall protects cells from environmental challenges; it is proposed that osmotic/mechanical stresses and cell wall integrity regulate the expression of cell wall genes in Chlamydomonas[60]. However, the underlying mechanisms of how the cell wall responds to high temperatures are largely under-explored. Cell walls form around daughter cells during cell division. Under diurnal regulation, many cell wall genes are up-regulated following the up-regulation of many cell cycle genes during cell division[53]. The expression pattern of cell wall genes in our data may be related to the inhibited and resumed cell cycles during and after heat treatments, respectively.

Heat at 35°C and 40°C induced unique transcriptional responses (Fig. 3a, Supplementary Fig. 4, 5, 6, 9, 10). Heat at 35°C induced a unique gene set that was not induced under 40°C, including genes involved in gluconeogenesis and glycolysis, mitochondrial assembly, and a putative calcium channel (Supplementary Fig. 5c–f, Supplementary Data 1, 3). While most of the overlapping DEGs between 35°C and 40°C treatments were more strongly differentially expressed at 40°C than at 35°C, a fraction of the overlapping DEGs displayed a larger fold-change at 35°C than at 40°C (Supplementary Fig. 6). One group of genes with higher upregulation at 35°C than at 40°C are those low-CO$_2$ inducible (LCI) genes, e.g., *LCI26* (at reaching high temperature and 8-h heat), *LCI19* (16-h heat) (Supplementary Data 1), suggesting an effort to compensate for increased CO$_2$ demands with increased growth at 35°C. Many of these uniquely regulated genes in the 35°C treatment have unknown functions and may include novel candidates important for acclimation to moderate high temperature.

Both 35°C and 40°C induced starch accumulation but possibly for different reasons (Fig. 9w, x, 10). The increased starch in 35°C treated cells may be due to increased acetate uptake/assimilation, gluconeogenesis and the glyoxylate cycle, as evidenced by an induction of proteins related to these pathways (Fig. 2f, Supplementary Fig. 8). In Chlamydomonas, acetate uptake feeds into the glyoxylate cycle and gluconeogenesis for starch biosynthesis[40,41]. The increased starch in 40°C treated cells may be due to inhibited cell division (Figs. 4 and 5), resulting in starch storage exceeding its usage[25,26]. Starch accumulation could also be an electron sink to alleviate the over-reduced electron transport chain during 40°C heat treatment[18]. Heat treated Arabidopsis plants (42°C for 7 h) also had increased starch[43]. Several genes involved in starch biosynthesis were induced during 40°C heat and early recovery (Supplementary Fig. 9f). The over-accumulated starch during 40°C may also contribute to the downregulation of genes involved in acetate uptake and assimilation (Supplementary Fig. 8g, h).

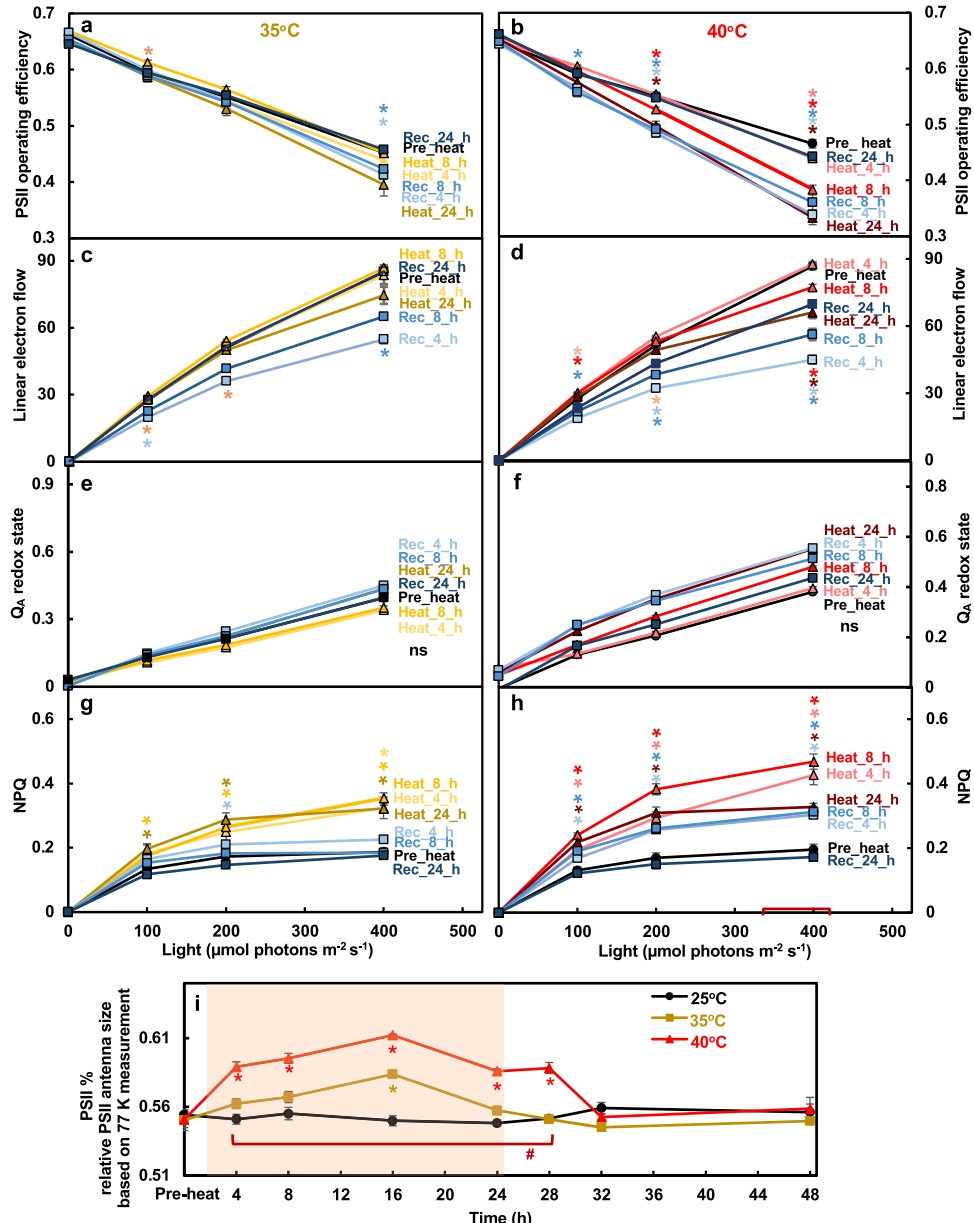

**Fig. 7 Heat of 40°C impaired PSII efficiency more than heat of 35°C while both induced NPQ.** Algal cultures harvested from PBRs before, during and after heat treatment at 35°C (**a, c, e, g, i**) or 40°C (**b, d, f, h, i**) were used for photosynthetic measurements. **a–h** Photosynthetic parameters measured using room temperature chlorophyll fluorescence. (**a, b**) PSII efficiency, the data at 0 μmol photons m$^{-2}$ s$^{-1}$ light are the maximum PSII efficiency in dark and data from light phase are PSII operating efficiency in light-adapted cells. **c, d** Linear electron flow, accounting for the changes of PSII antenna size during the treatments as in (**i**). **e, f** $Q_A$ redox state, the redox state of chloroplastic quinone A ($Q_A$), the primary electron acceptor downstream of PSII; the bigger number of $Q_A$ redox state means more reduced $Q_A$. **g, h** Nonphotochemical quenching, NPQ. **i** Relative PSII antenna fraction, percentage of light distributed to PSII measured by 77 K chlorophyll fluorescence. Mean ± SE, $n = 3$ biological replicates. Statistical analyses were performed using two-tailed t-test assuming unequal variance by comparing treated samples with the pre-heat samples under the same light (**a–h**, *) or constant 25°C samples at the same time point (**i**, *), or by comparing samples between 35°C and 40°C at the same time point (**i**, #). **a–i** P values were corrected by FDR. *, $p < 0.05$, the colors and positions of asterisks match the treatment conditions and time points, respectively. (**i**) #, $p < 0.05$, the positions of pound signs match the time points. **e, f** Not significant, ns.

Most of the reducing power from acetate assimilation is used in mitochondrial respiration[40,41]. However, the downregulated transcripts related to mitochondrial electron transport (Supplementary Fig. 15c), the heat sensitivity of mitochondrial respiration rates (Fig. 8h), and the over-accumulated starch may restrict acetate uptake and assimilation during 40°C heat treatment.

Heat at 35°C stimulated growth but 40°C decreased it (Fig. 1a, b). We quantified growth based on the rate of chlorophyll increase and medium consumption in PBRs under the turbidostatic mode,

monitored by OD$_{680}$ which is proportional to chlorophyll content (Supplementary Fig. 1a–c). Under our experimental conditions with little nutrient and light limitation, our results showed that cells exposed to 35°C reached the maximum OD$_{680}$ faster than cell exposed to 40°C or kept at constant 25°C. The stimulated growth in liquid cultures under 35°C was confirmed by growth on plates (Supplementary Fig. 2) and was consistent with increased transcripts and proteins related to mitochondrial electron transport as well as increased mitochondrial relative volume in 35°C-treated cells

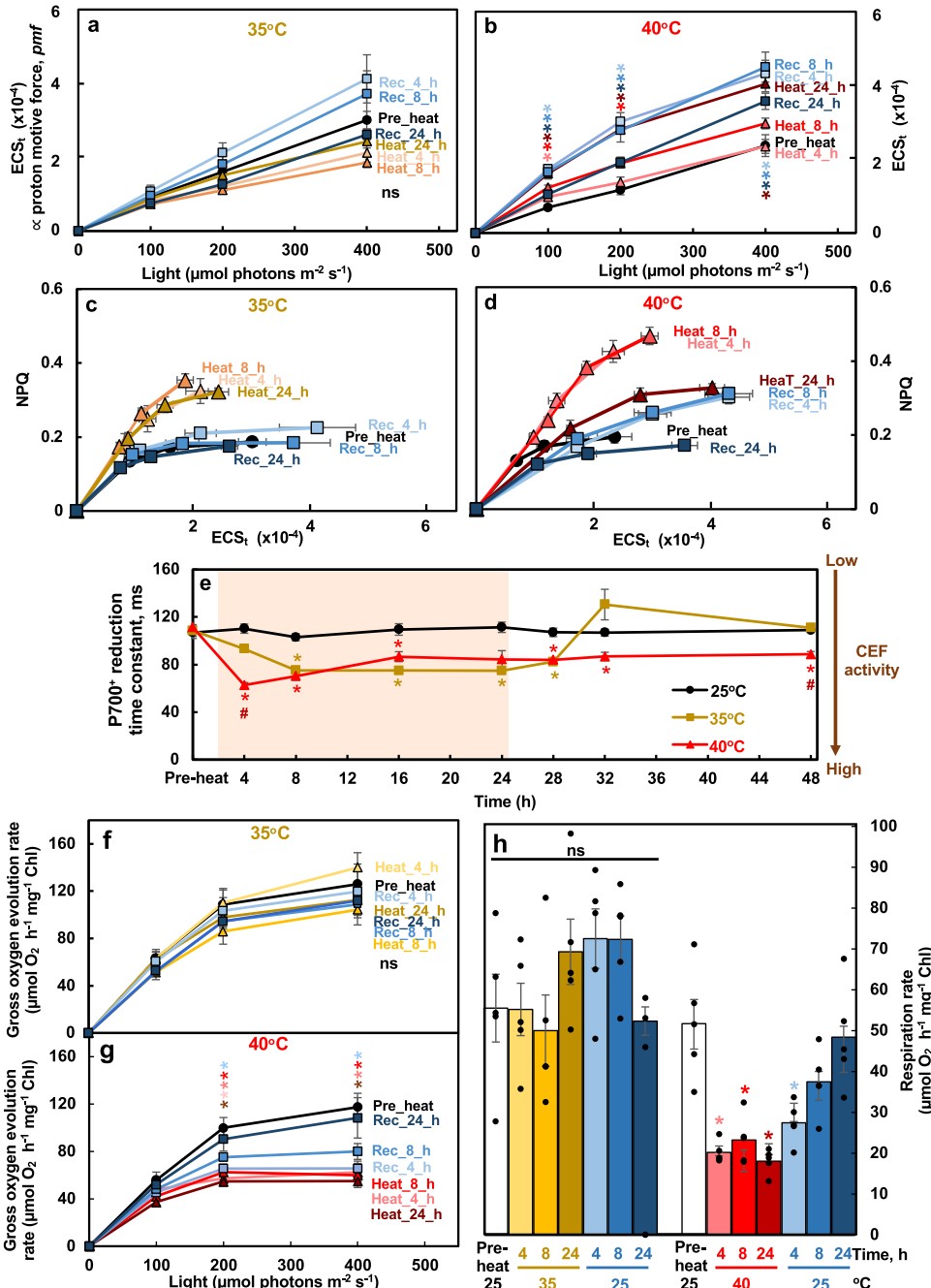

**Fig. 8 Heat at 40°C induced transthylakoid proton motive force, NPQ, CEF, but reduced O2 evolution and respiration rates in Chlamydomonas cells.**
**a**, **b** Heat treatment of 40°C increased the transthylakoid proton motive force (*pmf*). ECS$_t$, measured by electrochromic shift (ECS), represents the transthylakoid *pmf*. **c**, **d** NPQ was more sensitive to ECS$_t$ (or *pmf*) during both heat treatments, with higher NPQ produced at a given *pmf*. NPQ, non-photochemical quenching, measured using room temperature chlorophyll fluorescence. **e** Both 35°C and 40°C induced the activity of cyclic electron flow around PSI (CEF) although with different dynamics and reversibility. P700$^+$ reduction to measure CEF in the presence of 10 μmol DCMU to block PSII activity; the smaller P700$^+$ reduction time constant indicates faster P700$^+$ reduction and higher CEF activity. The red shaded area depicts the duration of the high temperature. **f**, **g**, **h** Gross O$_2$ evolution rates and respiration rates were reduced during the 40°C heat treatment, measured using a Hansatech Chlorolab 2 Clark-type oxygen electrode. Mean ± SE, $n = 5$ biological replicates. Statistical analyses were performed using two-tailed t-test assuming unequal variance by comparing treated samples with the pre-heat samples under the same light (**a**, **b**, **f**, **g**, **h**, *) or constant 25°C samples at the same time point (**e**, *), or by comparing samples between 35°C and 40°C at the same time point (**e**, #). **a**, **b**, **e**, **f**, **g** P values were corrected by FDR. *, $p < 0.05$, the colors and positions of asterisks match the treatment conditions and time points, respectively. (**e**) #, $p < 0.05$, the positions of pound signs match the time points. Not significant, ns.

(Supplementary Fig. 15). The increased protein levels in acetate uptake/assimilation and gluconeogenesis and glyoxylate cycles (Supplementary Fig. 8) may contribute to the faster growth under 35°C heat. Maize plants grown under moderate high temperature of 33°C had increased biomass but decreased biomass under higher temperature of 37°C as compared to controls at 31°C[61]. Similar temperature effects were also reported in synchronized algal cultures[62,63], consistent with our data. With increasing high

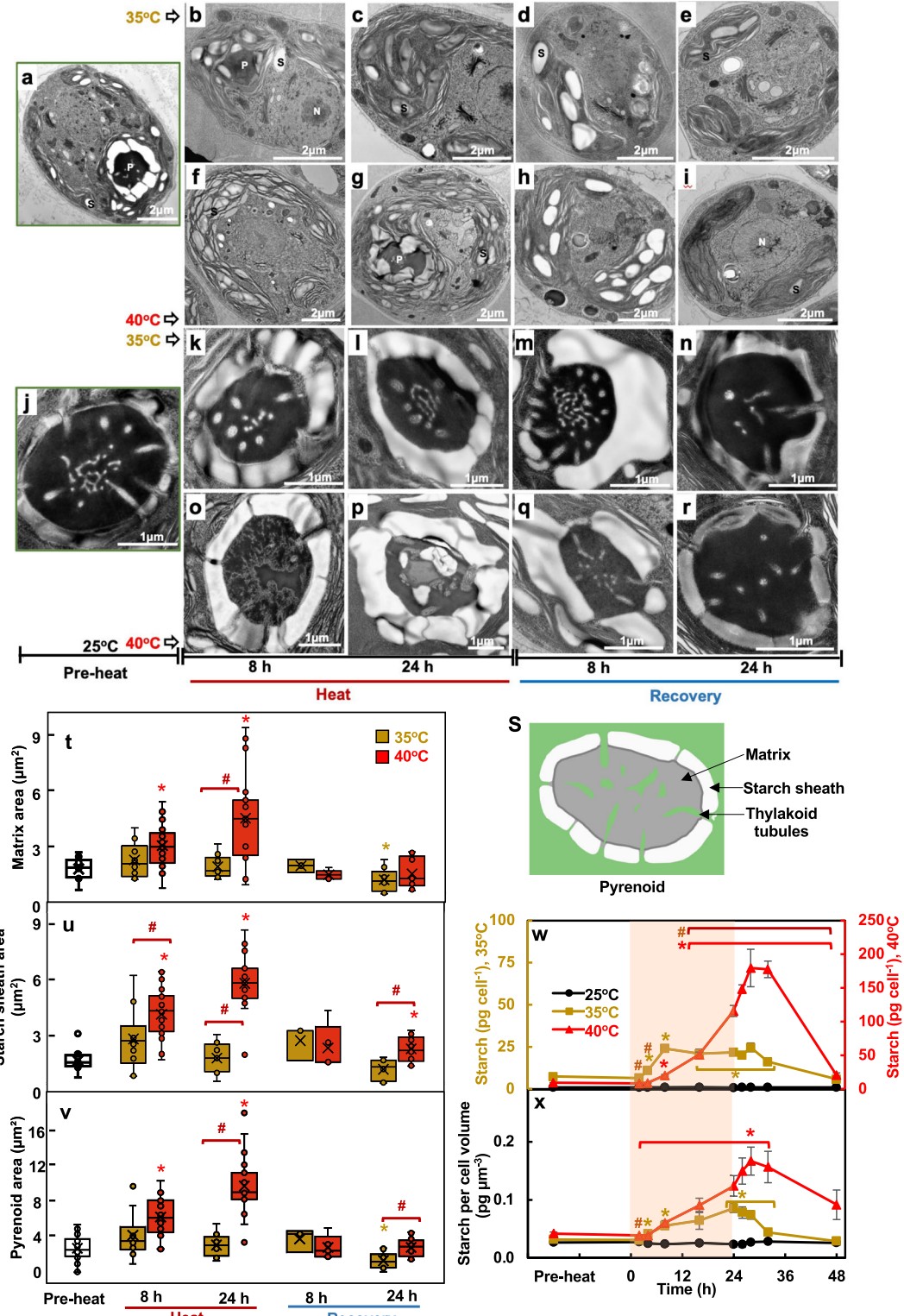

**Fig. 9 Only 40°C altered thylakoid and pyrenoid ultrastructure while both 35°C and 40°C treatments stimulated starch accumulation.**
**a–r** Representative transmission electron microscopy (TEM) images of algal cells or pyrenoids at different time points before (**a**, **j**), during, and after heat treatment at 35°C (**b–e**, **k–n**) or 40°C (**f–i**, **o–r**). **s** Cartoon representation of Chlamydomonas pyrenoid structure. **t**, **u**, **v** Areas of pyrenoid matrix, starch sheath and the whole pyrenoids, respectively, quantified using ImageJ and TEM images from algal samples harvested before, during, and after heat treatments of either 35°C (brown) or 40°C (red) at the indicated time points. The data is presented as boxplot based on Tukey-style whiskers. Median values are represented by the horizontal black lines and mean values by the X sign inside each rectangular box. **w**, **x** Starch quantification using starch assay kits. Values are mean ± SE, $n = 3$ biological replicates. The red shaded areas depict the duration of the high temperature. **t–x** Statistical analyses were performed using two-tailed t-tests assuming unequal variance by comparing treated samples with pre-heat (**t–v**) or 25°C at the same time point (w, x) (*, $p < 0.05$, the colors of asterisks match the treatment conditions), or by comparing samples between 35°C and 40°C at the same time point (#, $p < 0.05$). (**w**, **x**) $P$ values were corrected by FDR.

temperatures, the growth of photosynthetic organisms may accelerate first, then decrease when the high temperature exceeds a certain heat threshold.

The adaptive transcriptional changes in response to 40°C include rapid induction of transcripts encoding HSFs, HSPs, and ROS scavenging enzymes (Supplementary Fig. 9c, e, 10a). Chlamydomonas has two HSFs, HSF1 and HSF2[10]. HSF1 is a canonical HSF similar to plant class A HSFs and a key regulator of the stress response in Chlamydomonas[11], while the function of HSF2 is unclear. Our transcriptome data showed that both *HSF1* and *HSF2* were induced during early heat of 35°C and 40°C (Supplementary Fig. 10a), suggesting the potential role of HSF2 in heat regulation. Interestingly, *HSF1* was also induced during the early recovery phase after 40°C heat, possibly due to its potential roles in maintaining some heat responsive genes after heat treatment. HSF1 was shown to also be involved in altering chromatin structure for sustained gene expression[10,64]. HSP22E/F are small heat shock proteins targeted to the chloroplast, function in preventing aggregation of unfolded proteins, and are induced at temperatures at or above 39°C in Chlamydomonas[13,65]. The transcripts of *HSP22E/F* were induced transiently but strongly during 0.5- to 1-h heat of 40°C, but also during the first 4-h of recovery after 40°C heat (Supplementary Fig. 9c), suggesting their roles not only during heat but also the recovery from heat of 40°C.

Additionally, cells treated with 40°C heat had increased photoprotection (Figs. 7 and 8) and related transcripts were up-regulated (Supplementary Fig. 9a, e.g., *LHCSR*, *ELI*, and *PSBS*). With compromised photosynthesis under 40°C, the increased transthylakoid proton motive force (*pmf*), NPQ formation, sensitivity of NPQ to the *pmf*, and CEF activity were all helpful to dissipate excess light energy and reduce heat-induced oxidative stress. CEF generates only ATP but no NADPH, balances the ATP/NADPH ratio, contributes to the generation of *pmf*, and protects both PSI and PSII from photo-oxidative damage[66,67]. Increased CEF activity has been frequently reported under various stressful conditions in land plants[15,68,69] and algae[70–72]. It is proposed that the reduced plastoquinone (PQ) pool activates CEF in algae[72–74]. CEF is also proposed to provide the extra ATP needed for the carbon concentrating mechanisms (CCM) in Chlamydomonas[75]. Our results showed that increased CEF at 40°C (Fig. 8e) concurred with the induced transcripts involved in CCM (Supplementary Fig. 9b), and the increased proteins of PSI subunits (Fig. 6d).

Cells treated with 40°C heat had increased PSII/PSI ratio measured by 77 K chlorophyll fluorescence (Fig. 7i), as previously reported[18]. The 77 K chlorophyll fluorescence is often used to monitor the stoichiometries of PSII and PSI[76]. The ratios of PSII and PSI from the 77 K fluorescence emission is an indicator of the relative antenna size of each photosystem[77]. Hemme et al., (2014) reported that 42°C treated Chlamydomonas cells showed a blue shift of PSI emission peak from 713 to 710 nm, suggesting detachment of LHCI from PSI. In 40°C treated Chlamydomonas cells, we also observed the minor blue shift of the PSI emission peak but also an increased emission peak around 695 nm (Supplementary Fig. 12), which is associated with the PSII core antenna CP47[76]. Our spectral changes indicated reduced or detached PSI antenna but increased PSII antenna, thus an increased PSII/PSI ratio, suggesting relatively smaller antenna associated with PSI than PSII in cells treated with 40°C heat. Under high salt conditions, the Antarctic alga Chlamydomonas sp. UWO 241 forms a PSI-Cytochrome $b_6f$ supercomplex with constitutively high rates of CEF but absence of a discernible PSI peak in 77 K chlorophyll fluorescence emission[78–80]. Chlamydomonas forms the PSI-Cytochrome $b_6f$ supercomplex to facilitate CEF under anaerobic conditions[81,82]. Steinbeck et al. (2018) proposed that the dissociation of LHCA2/9 from PSI supported

the formation of the PSI-Cytochrome $b_6f$ supercomplex. In Chlamydomonas, the PSI core associates with LHCI which is comprised of ten LHCA subunits and LHCA2/9 are suggested to be weakly bound to the PSI core[83]. The PSI chlorophyll fluorescence under 77 K is mainly due to chlorophyll a in the LHCAs[76]. Combining our results with previous reports, we propose that heat-induced dissociation of LHCAs (possibly LHCA2/9) from the PSI core may facilitate the formation of the PSI-Cytochrome $b_6f$ supercomplex and increase CEF activity.

Cells treated with 40°C heat had altered pyrenoid structures (Fig. 9j, o, p). Algae utilize pyrenoids to concentrate $CO_2$ around Rubisco through the CCM[84,85]. In Chlamydomonas, pyrenoids consist of three major components: starch sheath (a diffusion barrier to slow $CO_2$ escape), pyrenoid matrix (Rubisco enrichment for $CO_2$ fixation), and thylakoid tubules (delivery of concentrated $CO_2$ and diffusion path of Calvin-Benson Cycle metabolites) (Fig. 9s)[86]. Several pyrenoid-localized proteins sharing a conserved Rubisco-binding motif are proposed to mediate the assembly of the pyrenoid in Chlamydomonas:[87] the linker protein Essential Pyrenoid Component 1 (EPYC1) links Rubisco to form the pyrenoid matrix;[88,89] the starch-binding protein Starch Granules Abnormal 1 (SAGA1) mediates interactions between the matrix and the surrounding starch sheath;[90] the thylakoid-tubule-localized transmembrane proteins RBMP1/2 mediate Rubisco binding to the thylakoid tubules in the pyrenoid[87]. From our TEM images, thylakoid tubules appeared to be absent from the pyrenoid matrix in cells treated with 40°C heat, which may suggest that 40°C heat disrupts the interaction of thylakoid tubules with the pyrenoid matrix and compromises CCM efficiency. The transcripts of *EPYC1*, *SAGA1*, and *RBMP2* were induced during 40°C heat (Supplementary Fig. 9b). We propose that 40°C heat may increase the disorder of the pyrenoid structure and Chlamydomonas cells compensate for this by inducing transcripts encoding the pyrenoid-structure-maintaining proteins mentioned above. Several other transcripts related to the CCM, e.g., low $CO_2$ inducible proteins, LCIA/D/E/C (helping maintaining $CO_2$ concentration in pyrenoids), were all up-regulated during 40°C heat (Supplementary Fig. 9b), which may suggest the attempt to maintain the CCM and compensate for the heat induced photorespiration[18] as well as $CO_2$ leakage from pyrenoids. *SAGA1* and *LCIA/D/E* were also induced during early recovery, suggesting the efforts to recover the CCM and/or coordinate pyrenoid division with cell division after 40°C heat. The increased CCM transcripts during 40°C heat and early recovery may be an adaptive response to alleviate the over-reduced electron transport chain.

The increased chlorophyll during 40°C heat may be a maladaptive response. Cells treated with 40°C heat had more than 4x increased chlorophyll per cell (Fig. 5d), which could not be fully explained by increased cell volume (Supplementary Fig. 11a). Increased chlorophyll in heat treated Chlamydomonas cells has been reported previously[18], but the underlying mechanisms are unclear. Heat at 40°C appeared to promote chlorophyll biosynthesis. The gene encoding the key chlorophyll synthesis enzyme, porphobilinogen deaminase, *PBGD2*[91], was upregulated during 40°C heat (Supplementary Fig. 9d). Considering the compromised photosynthesis and decreased growth during 40°C heat, increasing chlorophyll levels to this extent is toxic. The elevated chlorophyll may lead to increased light harvesting with decreased photosynthesis in 40°C treated cells, resulting in ROS production. Chlorophyll contents positively correlate with nitrogen availability[92] and we found many genes related to the nitrogen assimilation pathways were up-regulated during 40°C heat (Supplementary Fig. 9g), providing a possible explanation for increased chlorophyll during 40°C heat. Maize plants showed greater sensitivity to high temperatures with increased

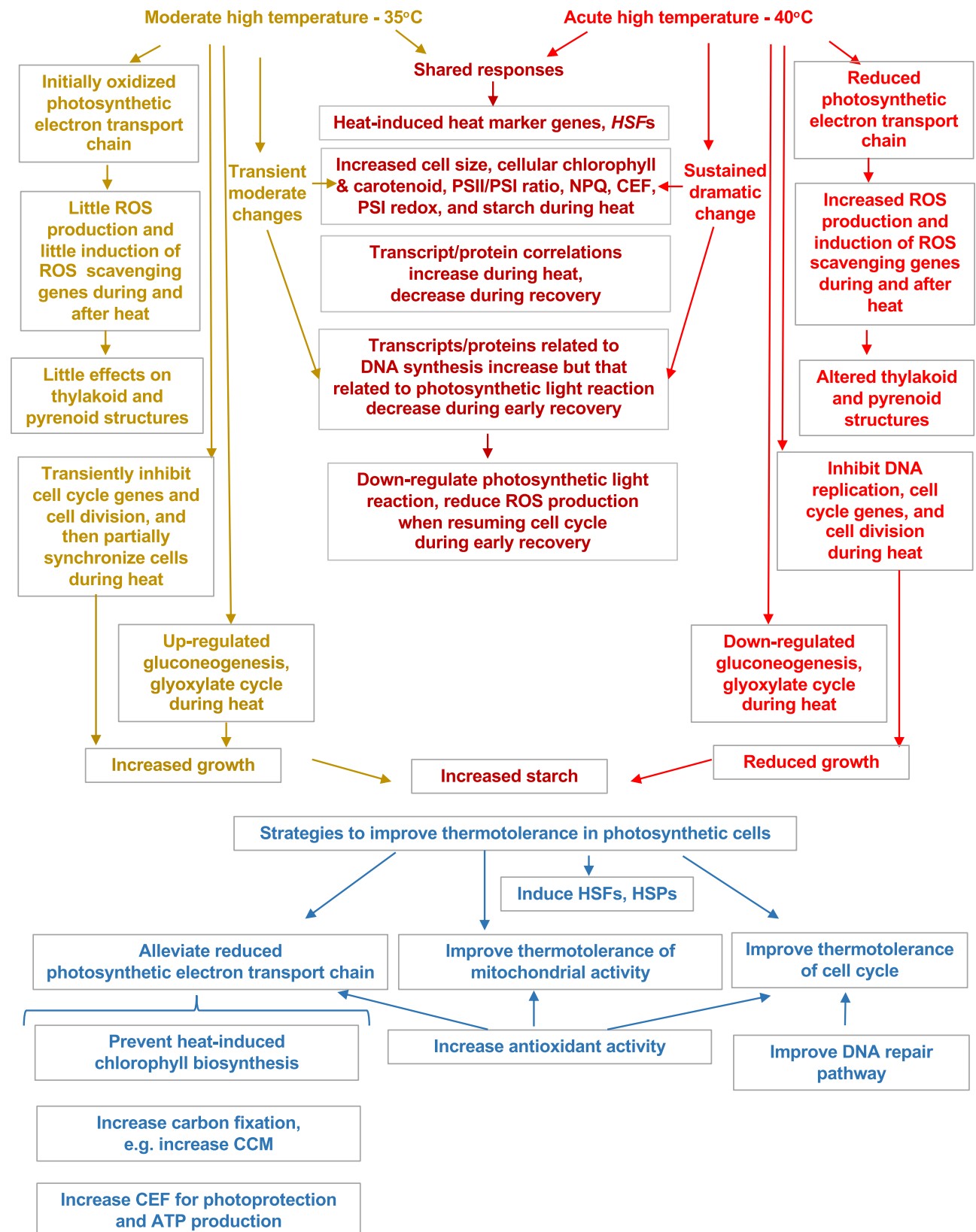

**Fig. 10 Chlamydomonas has shared and unique responses to moderate and acute high temperatures of 35°C and 40°C. Top panel**: Summarized results from multiple-level anlayses revealed the unique responses to 35°C (left, brown), unique to 40°C (right, bright red), and those shared between the two treatments (middle, dark red). **Bottom panel** (blue): Our results can be used to inform strategies to improve thermotolerance in photosynthetic cells.

nitrogen fertilization[93], which may support the possible links among nitrogen assimilation, chlorophyll biosynthesis, and heat responses. In land plants, long-term (e.g. several days) heat stress reduces chlorophyll content[94,95], however, the underlying mechanisms by which chlorophyll is degraded during long-term heat remain elusive[94]. Chlamydomonas cells treated at 39°C for more than one day had initially increased chlorophyll (8~16-h heat) followed by chlorophyll loss, cell bleaching, and death (33-h heat)[25]. It is possible that preventing chlorophyll increase during acute high temperature (especially early stage) could lead to improved thermotolerance in algae.

Combining our systems-wide analyses, we could distinguish adaptive versus maladaptive heat responses as mentioned above. The potential engineering targets for improved thermotolerance may include these adaptive heat responses, e.g., heat induced HSFs, HSPs, photoprotection, CEF, antioxidant pathways, and CCM transcripts (Fig. 10). The maladaptive responses could also be the targets for improved thermotolerance if we could find mediating solutions to reduce these changes, e.g., heat-induced chlorophyll. The cell cycle arrest induced by acute high temperature may be maladaptive but also adaptive: the halted cell division may be one of the main reasons for over-accumulated starch, reduced photosynthetic electron transport chain, and increased ROS production; on the other side, cell cycle arrest under acute high temperature may prevent damages/errors during DNA replication when DNA repair pathways are compromised by heat[58]. Thus, increasing thermotolerance of the DNA repair and cell cycle pathways may also be strategies to improve heat tolerance in photosynthetic cells. Furthermore, mitochondrial activity was stimulated slightly by 35°C heat but sensitive to 40°C heat (Fig. 8 h, Supplementary Fig. 15), suggesting mitochondrial activity could be another target to improve thermotolerance. Additionally, the genes associated with TM1 with early heat induction may include pathways that are essential for heat tolerance (Fig. 3a, Supplementary Data 5). Finally, we compared our algal heat transcriptome with that in Arabidopsis (heat at 42°C for 7 h)[43] and identified a set of highly conserved heat-induced genes sets (Supplementary Data 1), which may provide potential targets to improve heat tolerance in land plants.

In summary, Chlamydomonas is an excellent model to study the heat response and its regulation at the cellular level in photosynthetic cells. Our research helped fill the knowledge gaps regarding how algae respond to and recover from different intensities of high temperatures at multiple levels, discovered the increased transcript/protein correlation during heat treatments, showed the dynamics of photosynthesis in response to high temperatures, and revealed the antagonistic interaction between DNA replication and photosynthetic light reactions during the recovery from both moderate and acute high temperatures. Through systems-wide analyses, we advanced our understanding of algal heat responses and identified engineering targets to improve thermotolerance in green algae and land plants.

## Methods

**Strains and culture conditions**. *Chlamydomonas reinhardtii* wildtype strain CC-1690 (also called *21gr*, *mt+*, from the Chlamydomonas resource center)[96–99] was used in all experiments. CC-1690 were grown in standard Tris-acetate-phosphate (TAP) medium in 400 mL photobioreactors (PBRs) (Photon System Instruments, FMT 150/400-RB). Cultures were illuminated with constant 100 μmol photons m² s⁻¹ light (50% red: 50% blue), mixed by bubbling with filtered air at a flow rate of 1 L/min. After PBR inoculation at initial cell density of $0.5 \times 10^6$ cells/mL, cultures were allowed to grow to a target cell density of $2.00 \times 10^6$ cells/mL corresponding to around 4.0 μg/mL chlorophyll content in log-phase growth at 25°C. Then, the target cell density was maintained turbidostatically using $OD_{680}$ by allowing the culture to grow to 8% above the target cell density before being diluted to 8% below the target cell density with fresh TAP medium provided through peristaltic pumps. Through the turbidostatic mode, the PBR cultures had exponential growth between dilution events. The $OD_{680}$ measurement during exponential growth phases in between dilution events was $\log_2$

transformed, and the relative growth rate was calculated using the slope of $\log_2(OD_{680})$, while the inverse of the slope yielded the doubling time of the culture (Supplementary Fig. 1a, b). All algal liquid cultivation used in this paper was conducted in PBRs with the conditions mentioned above.

**High-temperature treatments in PBRs**. Algal cultures in PBRs were maintained turbidostatically using $OD_{680}$ for 4 days at 25°C to allow cultures to adapt to steady growth conditions before heat treatments (Fig. 1b). PBR temperatures were then shifted to moderate or acute high temperature conditions (35°C or 40°C in different PBRs) for 24 h, then shifted back to 25°C for 48 h for recovery. PBR cultures grown under constant 25°C served as controls. Cultures were maintained turbidostatically during the entire experiment and harvested at different time points for various measurements. Cell density and mean cell diameter were measured using a Coulter Counter (Multisizer 3, Beckman Counter, Brea, CA). For the data in Fig. 1a, algal cultures were maintained in PBRs at 25°C for 4 days before switching to 30°C, 35°C, or 40°C for 2 days (different temperature switches in separate PBRs). The relative growth rates were calculated at the end of 2-day treatment of each temperature.

**Spotting test for cell viability and growth**. Cultures harvested from PBRs were diluted to $2 \times 10^4$ cells mL⁻¹ or $1 \times 10^5$ cells mL⁻¹ with TAP medium and 10 μL aliquots of the diluted cultures were spotted on 1.5% TAP agar plates and grown in temperature-controlled incubators under 25°C or 35°C with constant white LED light of 150 μmol photons m⁻² s⁻¹ for 44 h or 3 days. After 44-h growth, algal spots with 200 cells were imaged by a dissecting Leica microscopy and were used for growth quantification. Colony number and area were quantified using ImageJ. Viability was calculated as the number of colonies on plates divided by the number of cells spotted. Algal spots with 200 and 1000 cells were imaged after 3-day-growth for visual representations.

**High temperature treatments in water bath**. To measure the effects of heating speed on cell viability (Supplementary Fig. 1f), control PBR cultures without heat treatments were incubated in a water bath. Gradual heat treatment from 25°C to 41°C took place over 25 min, then cultures were kept at 41°C for 2 h. Directly heated samples were incubated in a water bath which was pre-heated to 41°C then kept at 41°C for 2 h (sharp temperature switch). Cell viabilities after 2-h 41°C heat treatment (either gradual or sharp heating) were assayed using the spotting test as above. Because PBRs cannot switch from the control to high temperatures in less 25 min, a water bath was used for this test, as previously reported[13,18,33].

**RNA extraction and RT-qPCR**. At each time point, 2 mL PBR cultures were pelleted with Tween-20 (0.005%, v/v) by centrifugation at 1,100 x g and 4°C for 2 min. The cell pellet was flash frozen in liquid nitrogen and stored at −80°C before processing. Total RNA was extracted with the TRIzol reagent (Thermo Fisher Scientific, Cat No. 15596026) as described before with some modifications[53]. RNA was purified by RNeasy mini-column (Qiagen, Cat No. 74106) after on column digestion with RNase-free DNase (Qiagen, Cat No. 79256) according to the manufacturer's instructions. RNA was quantified with Qubit™ RNA BR Assay Kit, (Life technology, Cat No. Q10210). Total 0.4 μg RNA was reverse transcribed with oligo dT primers using SuperScript® III First-Strand Synthesis System (Life technology, Cat No. 18080-051) according to the manufacturer's instructions. Quantitative real-time PCR (RT-qPCR) analysis was carried out using a CFX384 Real-Time System (C 1000 Touch Thermal Cycler, Bio-Rad, Hercules, California) using SensiFAST SYBR No-ROS kit (Bioline, BIO-98020). PCR was set up as follows: (1) 2 min at 95°C; (2) 40 cycles of 5 s at 95°C, 10 s at 60°C and 15 s at 72°C; (3) final melt curve at 60°C for 60 s, followed by continuous ramping of temperature to 99°C at a rate of 0.5°C s⁻¹. Melting curves and qPCR products was checked after PCR cycles to ensure there are no primer dimers or unspecific PCR products. All qPCR products were sequenced to verify their identifies. Expression of G-protein β-subunit-like polypeptide CBLP (Cre06.g278222) (Schloss, 1990) remain stable among all time points, and were used as internal controls[100]. The relative gene expressions were calculated relative to the gene's own expression in pre-heat by using the $2^{-\Delta\Delta CT}$ method as described previously[101–103]. Three biological replicates for each time point and treatment were conducted. The qPCR primers used are listed in Supplementary Table 2.

**Transcriptomics**. RNA libraries were prepared and sequenced by the Joint Genome Institute (JGI, Community Science Program) using the NovaSeq platform and generated 150-nt paired-end reads, with the goal of 20 million genome-mappable reads per sample. Samples were quality control filtered using the JGI BBDuk and BBMap pipelines[104]. BBDuk trims adapter sequences, and quality trims reads with low complexity, low quality scores, and reads with 1 or more "N" bases. BBMap removes reads that map to common contaminant genomes. Samples were quality assessed using FastQC[105] and mapped to the *Chlamydomonas reinhardtii* v5.6 genome[106] using STAR[107,108] with the maximum number of mapped loci set to 1 and the maximum mismatches per read set to 1, resulting in >92% of all reads being uniquely mapped to the *Chlamydomonas reinhardtii* v5.6 genome. Reads per feature were counted via HT-seq count[109]. The dataset was filtered for genes that met minimum read count cutoffs of at least 10 mapped reads in at least 10% of the

samples, resulting in 15,541 genes for downstream analyses. Two-dimensional Uniform Manifold Approximation and Projection (UMAP) was used to reveal clusters of RNA-seq data[110].

Differential expression modeling was performed on transcripts per million (TPM) normalized read counts using a generalized linear mixed-effect model and a negative binomial distribution. Treatment time points were compared to pre-heat controls. Significant differential expression was defined as $|\log_2$ fold-change$| > 1$, FDR $< 0.05$, and $|$(control mean TPM)—(treatment mean TPM)$| \geq 1$. The sign of $|$ means absolute values. FDR correction was performed using the Benjamini-Hochberg method[111]. Heatmaps were generated using the R package pheatmap (version 1.0.12. https://CRAN.R-project.org/package=pheatmap). Weighted correlation network analysis (WGCNA) was performed on TPM normalized read counts that met minimum read count cutoffs[112]. Expression patterns from 35°C and 40°C were modeled together as a signed network, requiring at least 50 genes per module and combining modules with similarity >0.25. All genes were tested against all eigengene modules using ANOVA (FDR < 0.05) and were assigned to the module with highest significance. Functional enrichment analysis was performed using hypergeometric tests with subsequent FDR control based on MapMan annotations for Chlamydomonas reinhardtii (FDR < 0.05)[111]. Surprisal analysis was applied to dissect the transcriptional regulatory response into its major components[113]. Constraint potentials were calculated for each transcriptional data matrix, whose signals determine the deviations of the transcriptome from a theoretical balance state of minimal free energy[114]. The overall importance of the transcriptional pattern was estimated from its amplitude and decreases with increasing constraint index. Accordingly, the three major constrains were kept.

**Proteomics**. Algal samples (2 mL) harvested from PBRs were centrifuged to remove supernatant, flash frozen in liquid nitrogen, and stored at −80°C until use. Proteins were extracted using the IST sample preparation kit (PreOmics GmbH, Germany). Cell pellets (about 200 µg protein) were lysed in 50 µL lysis buffer, and 20% of the lysate was added into 40 µL lysis buffer and sheared in a sonicator (VWR Aquasonic 250D, 35Khz) for 3 min, then digested in a heating block at 37°C with gentle shaking for 3 h followed by stopping the digestion with the stop buffer. Peptides were purified using the PreOmics cartridges according to the manufacturer's instruction. Eluted peptides were transferred to 96 well plates, dried under vacuum for 2 h, and then dissolved in 30 µL of LC-loading buffer included in the kit. Finally, 5 µL of the suspension was used for LC-MS/MS analysis.

LC-MS/MS (Liquid chromatography–mass spectrometry) was carried out on an Orbitrap Fusion Lumos (Thermo Fisher Scientific, San Jose, CA) mass spectrometer coupled with a U3000 RSLCnano HPLC (Thermo Fisher Scientific, San Jose, CA). The peptide separation was carried out on a C18 column (Fritted Glass Column, 25 cm × 75 µm, Reprosil-Pur 120 C18-AQ, 1.9 µm, made by ESI Source Solution, LLC., Woburn, MA) at a flow rate of 0.3 µL/min and the following gradient: Time = 0–4 min, 2% B isocratic; 4–8 min, 2–10% B; 8–83 min, 10–25% B; 83–97 min, 25–50% B; 97–105 min, 50–98%. Mobile phase consisted of A, 0.1% formic acid; mobile phase B, 0.1% formic acid in acetonitrile. The instrument was operated in the data-dependent acquisition mode in which each MS1 scan was followed by Higher-energy collisional dissociation (HCD) of as many precursor ions in 2 second cycle (Top Speed method). The mass range for the MS1 done using the FTMS was 365 to 1800 m/z with resolving power set to 60,000 @ 200 m/z and the automatic gain control (AGC) target set to 1,000,000 ions with a maximum fill time of 100 ms. The selected precursors were fragmented in the ion trap using an isolation window of 1.5 m/z, an AGC target value of 10,000 ions, a maximum fill time of 100 ms, a normalized collision energy of 35 and activation time of 30 ms. Dynamic exclusion was performed with a repeat count of 1, exclusion duration of 30 s, and a minimum MS ion count for triggering MS/MS set to 5000 counts.

**Protein identification and quantification**. Quantitative analysis of MS measurements was performed using MaxQuant 1.6.12.0[115]. The library used to perform peptide spectrum matching was created based on version JGI5.5 of the Chlamydomonas reinhardtii genome. The search space was extended including methionine oxidation and acetylation of protein N-termini as variable modifications. The false discovery rate (FDR) thresholds for peptide spectrum matching and protein identification were set to 1%. Protein abundances were estimated using the Label free Quantification (LFQ) algorithm[116].

**Data normalization and protein-level missing value imputation**. Normalization and missing value imputation were performed independently for 35°C and 40°C time courses. Proteins were excluded from the data set if there was no biological replicate group with more than one value among different time points, as these proteins were considered not suitable for quantitative downstream analysis. Following normalization using the median-of-ratios method[117], we used global variance estimates and local gene wise mean estimates to impute missing data points as independent draws from normal distributions. If a protein showed no quantification in a group of biological replicates at one-time point, it was checked if the protein was present in the adjacent time points (the time point before and after the one in query). If this was the case, the protein mean was imputed using k-Nearest-Neighbour imputation, followed by sampling from a normal distribution. If

adjacent time points had no values, no imputation was performed for the time point in the query. All further protein analyses were based on imputed values.

**Network generation**. All protein groups resulting from non-proteotypic peptides were duplicated to singletons and the intensities of each protein were log-transformed. To ignore proteins with constant abundance signals, they were filtered for significance by one-way ANOVA ($p < 0.05$). To generate a correlation matrix, the Pearson correlation coefficient was used. An absolute correlation threshold was determined by random matrix theory[118]. These thresholds were determined to be $\rho = 0.8194$ and $\rho = 0.8675$ for 35°C and 40°C, respectively. After filtering the correlation matrix accordingly, the nodes and edges were isolated and visualized in Gephi (www.gephi.org) using ForceAtlas2[119] and built-in modularity determination[120].

**Statistical testing of proteins**. Proteins that were identified by ambiguous (non-proteotypic) peptides or had missing values in at least one replicate were not considered for statistical testing. All time points were tested for significant accumulation or depletion in respect to a control measured prior to the start of the heat treatment. Dunnett's multiple-comparison test was applied with alpha levels of 0.05 and 0.01.

**Proteomics enrichment analyses**. To investigate the module compositions a term enrichment based on MapMan ontology was performed. The ontology tree of each term was expanded, so every protein could exist multiple times, corresponding to the number of ontology term levels. A hypergeometric test for each functional term was applied with subsequent FDR control by the Benjamini-Hochberg method[111]. To derive a representative signal shape of all proteins included in a module, its eigenvector was calculated based on complete time series, which were log-transformed and centered to an intensity of zero mean and unit variance[121]. For negatively correlating proteins, its corresponding eigenvector was inverted. Additional term enrichments were applied following the previous schema to gain insights into functions of positively or negatively correlating proteins of each module.

**Correlation of transcripts and proteins**. The $\log_2$(fold-change) of transcript reads and protein abundance were calculated in respect to the pre-heat samples. All available transcript/protein pairs were taken into consideration for correlation analysis. The heat period (HS) as well as the recovery period (RE) were split up into three windows each (HS: 0–1 h, 2–8 h, 16–24 h during the heat period; RE: 0–2 h, 4–8 h, 24–48 h during the recovery period after heat treatment). The average $\log_2$(fold-change) is determined for each window. Every identifier, that had both a transcript and protein associated with it, resulted in a transcript-protein fold-change pair for each window. By collecting all identifiers that are associated with a respective functional term a scatter plot of transcript-protein fold-change pairs were generated, and the Pearson correlation coefficient was calculated. By collecting the correlation coefficient for every present functional term, a density plot for each of the six windows was created. The resulting correlation coefficient histograms were smoothed using Silvermans rule of thumb for kernel density estimation[122]. See Supplementary Fig. 7 for the illustration.

All computational analyses on protein intensities were conducted using the open-source F# libraries FSharp.Stats, BioFSharp, and Plotly.NET. Linear regression, Benjamini–Hochberg correction, smoothing, correlation measures, and eigenvector calculations were performed using the FSharp.Stats version 0.4.1-beta. For ontology annotation and GSEA based on hypergeometric tests, we used BioFSharp version 2.0.0-beta4. Visualization of transcript-protein correlation was performed using Plotly.NET version 2.0.0-alpha5.

**DNA content and ploidy**. DNA content was analyzed by FACS (fluorescence-activated cell sorting) with modified protocol[123]. Cell cultures (10 mL) were collected and fixed in 30 mL ethanol:acetic acid (3:1) for 15 min at room temperature. Cells were spun down at room temperature at 4000 x g for 1 min and washed once with 1 mL phosphate-buffered saline (PBS). Then cells were collected by centrifugation, resuspended in 2 mL PBS with RNase A (100 µg/mL) for 2 h at 37°C, centrifuged again and finally resuspended in 2 mL PBS + 500 nM Sytox Green (Thermo Fisher Scientific, S7020). FACS was performed on an Accuri C6 instrument (BD Biosciences, Franklin Lakes, NJ), reading 20,000 cells per sample in the FL1 channel (488 nm exciting laser; emission filter: 530 ± 15 nm). A 90% attenuator was used to reduce signal below saturation levels. Data were analyzed with FlowJo software (BD Biosciences). Assignment of cell populations representing 1 C, or >1 C DNA content was determined. The raw fluorescence signal of 2 C was not two-times the signal at 1 C because there was background staining, which was cell-size-dependent[123]. Therefore, high-ploidy cells (which did not get much bigger during a multiple fission cycle) had DNA signal that was about proportional to actual DNA contents, while in low-ploidy cells the background contributed more. Thus, in the population of pre-heat samples, background is about $0.5 \times 10^5$ of the DNA content fluorescence signal (x axis). Before background subtraction, 1 C, 2 C, 4 C cells peaked at $3 \times 10^5$, $5.5 \times 10^5$, $10 \times 10^5$ of DNA content fluorescence signal, respectively. After background subtraction, 1 C, 2 C, 4 C cells were at $2.5 \times 10^5$, $5 \times 10^5$, $9.5 \times 10^5$, respectively. There was also a small peak at $20 \times 10^5$ which was

8 C. At 24 h heat of 40°C, cell size was about 2x bigger (Fig. 5a, b, c), with background of around 2 ×$10^5$, causing the right shift of the 1 C peak (Fig. 4k).

**Cell imaging using light microscopy**. Cultures harvested at select time points were fixed with 0.2% glutaraldehyde (VWR, Cat No. 76177-346). Cells were imaged with a Leica DMI6000 B microscope and a 63x (NA1.4) oil-immersion objective. Images shown are representative of results from at least three independent experiments (Fig. 5a).

**Pigment analysis**. Three biological replicates (each with two technical replicates) of 1 mL of PBR cultures were harvested at different time points, mixed with 2.5 µL 2% Tween20 (Sigma, P9416-100ML) to help cell pelleting, centrifuged at 18,407 g at 4°C, and stored in −80°C after removal of supernatant. Cell pellets were later thawed, resuspended in 1 mL of HPLC grade methanol (100%, Sigma, 34860-4L-R), vortexed for 1 min, incubated in the dark for 5 min at 4°C, and centrifuged at 15,000 g at 4°C for 5 min. Supernatant containing pigments was analyzed at 470, 652, 665 nm in a spectrophotometer (IMPLEN Nonophotometer P300) for carotenoids and chlorophyll a/b concentrations in µg mL$^{-1}$ using the following equations: Chl a + Chl b = 22.12*$A_{652}$ + 2.71*$A_{665}$, Chl a = 16.29*A665 − 8.54*A652, and Chl b = 30.66*A652 − 13.58*A665[124], and carotenoids = (1000* A470 − 2.86*Chl a − 129.2*Chl b)/221[125].

**Protein concentration and ROS determination**. Frozen sample pellets were thawed on ice and lysed by sonication in 10 mM Tris-HCl buffer[126]. Protein concentrations were determined by the method of Lowry (Lowry et al. 1951). The level of ROS was determined by the method described before with some modifications[127]. A 40 µg total protein extract was used for ROS measurement. Each sample was aliquoted and one of them was added with ascorbate (Thomas Scientific LLC, C988F55) to a final concentration of 100 mM. The samples containing ascorbate were used as background signal and subtracted from each experimental value later. The samples were incubated for 15 min at 25°C. The ROS indicator CM-H2DCFDA (Life Technologies, C6827) dissolved in 20% (v/v) DMSO was then added to a final concentration of 10 µM and incubated for 30 min at 30°C. The samples were transferred to 96 well microplates and ROS-related fluorescence was measured using Tecan Microplate reader M200 PRO with excitation at 485 nm and emission at 525 nm. The results were obtained from three biological replicates. Relative fold-change of ROS signal (compared to pre-heat) was either normalized to cell number or cell volume. Each of the three biological replicates included two independent measurements with two technical replicates.

**Spectroscopic measurement of photosynthetic parameters**. Photosynthetic measurements (chlorophyll fluorescence, electrochromic shift, and P700) were conducted using a multi-wavelength kinetic spectrophotometer/fluorometer with a stirring enabled cuvette holder (standard 1 cm pathlength) designed and assembled by the laboratory of Dr. David Kramer at Michigan State University using the method described before with some modifications[75]. A 2.5 mL volume (around 12~13 µg chlorophyll) of algal cells were sampled from the photobioreactors, supplemented with 25 µL of fresh 0.5 M NaHCO₃, loaded into a fluorometry cuvette (C0918, Sigma-Aldrich), and dark-adapted with a 10-min exposure to far-red light with peak emission of 730 nm at ~35 µmol photons m$^{-2}$ s$^{-1}$. All photosynthetic measurements were performed at the room temperature at 25°C for algal cultures sampled at different time points with different temperature treatments to investigate how the changes induced by high temperatures affected algal photosynthetic performance by comparing with control algal cultures treated and measured at 25°C. The maximum efficiency of PSII ($F_v/F_m$) was measured with the application of a saturating pulse of actinic light with peak emission of 625 nm at the end of the dark adaptation period and after turning off the far-red light. Our tests indicated that far-red illumination could increase the values of $F_v/F_m$ in dark-adapted algal cells. Far-red light was turned off during all chlorophyll fluorescence measurements to prevent its effect on chlorophyll fluorescence signals. Fluorescence measurements were taken with measuring pulses of 100 µs duration. The pulsed measuring beam was provided by a 505 nm peak emission (light emitting diode) LED filtered through a BG18 (Edmund Optics) color glass filter. After dark-adaptation, the algal sample was illuminated by a pair of LEDs (Luxeon III LXHL-PD09, Philips) with maximal emission at 620 nm, directed toward both sides of the cuvette, perpendicular to the measuring beam. Subsequent chlorophyll fluorescence and dark interval relaxation kinetic (DIRK) measurements were taken after 7.5-min adaptation of sequentially increasing light intensities of 100, 200, and 400 µmol photons m$^{-2}$ s$^{-1}$. DIRK traces measure the electrochromic shift (ECS). ECS results from light-dark-transition induced electric field effects on carotenoid absorbance bands[51,128] and is a useful tool to monitor proton fluxes and the transthylakoid proton motive force (*pmf*) in vivo[129,130]. ECS measurements were taken at each light intensity following the chlorophyll fluorescence measurements. DIRK traces were run with measuring beam of peak 520 nm and pulse duration of 14 µs capturing absorption changes from a light-dark-light cycle with each phase lasting 300 ms. Three near-simultaneously DIRK traces were averaged for one measurement.

**P700 measurement**. Algal cultures were sampled directly from PBRs at different time points with different treatments, supplemented with 0.5 M NaHCO₃, incubated with 10 µM DCMU (3-(3,4-Dichlorophenyl)−1,1-dimethylurea, Sigma, D2425) to block PSII activity, and dark adapted for 5 min before the measurement. PSI redox kinetics were monitored using a 703 nm measuring beam pulsed at 10-ms intervals using the spectrophotometer mentioned above as described previously with some modifications[75]. After a 5 s baseline measurement in the dark, samples were exposed to 720 nm far-red light at an intensity of 30 µmol photons m$^{-2}$ s$^{-1}$ for 5 s to preferentially oxidize PSI before monitoring the reduction of oxidized P700 (P700$^+$) in dark for 5 s. Addition of saturating flash at the end of far-red illumination did not change the amplitude of P700$^+$ and the reduction time constant of P700$^+$ significantly so the saturating flash was skipped in the finalized P700 measurement. Five measurements were averaged into one trace, and the reduction time constant of PSI was calculated by fitting to a first order exponential function to the reduction phase of the averaged trace. The reduction time constant of P700$^+$ in the absence of PSII activity (with DCMU) can be used to estimate the activity of cyclic electron flow around PSI (CEF)[74].

**77 K chlorophyll fluorescence measurement**. Chlorophyll fluorescence emission spectra were monitored at 77 K to estimate antenna sizes[76,131]. Sampled algal cultures from PBRs were immediately loaded into NMR tubes (VWR, cat. No. 16004-860) and frozen in liquid nitrogen. While still submerged in liquid nitrogen, frozen samples were exposed to excitation LED light with peak emission at 430 nm through a bifurcated fiber optic cable coupled to an LED light source[75]. Components were held in alignment using a 3D printed device. Chlorophyll emission spectra were recorded on an Ocean Optics spectrometer (cat. No. OCEAN-HDX-XR), and three consecutive traces were averaged into one measurement. Further dilution of the PBR samples gave identical fluorescence emission peak distributions, indicating little distortion of the signals by reabsorption. Spectral data were normalized to the PSII spectral maximum value at 686 nm, and the relative PSII antenna size (PSII%) was calculated using the normalized PSII peak and PSI peak (714 nm) using the formula PSII% = normalized PSII peak / (normalized PSII peak + normalized PSI peak).

See Supplementary Table 3 for formulas to calculate photosynthetic parameters. All measurements were taken with at least three biological replicates. Statistical significance was assessed using two-tailed t-test assuming unequal variance. For more than 20 comparisons, FDR using Benjamin-Hochberg correction method was performed with adjusted *p* value < 0.05 as significance.

**Oxygen evolution measurements**. Oxygen evolution was measured with Hansatech Chlorolab 2 based on a Clark-type oxygen electrode at room temperature at 25°C, following the method described before with some modifications[132]. Two-mL of cells (around 10 µg chlorophyll) supplemented with 20 µL of 0.5 M NaHCO₃ were incubated in the dark for 10 min. The dark respiration rate was measured at the end of the dark incubation. The rate of oxygen evolution was measured at increasing light intensities (100, 200 and 400 µmol photons m$^{-2}$ s$^{-1}$). Each light intensity lasted 5 min followed by 2 min dark. The rates of net oxygen evolution at each light intensity step were recorded for 1 min before the end of each light phase. Respiration rates were measured during each subsequent dark phase. For each time point, the respiration rates did not vary significantly after 10-min dark-adaptation or after different light intensities so only the respiration rates after 10-min dark-adaptation were plotted in Fig. 8h. The gross oxygen evolution rates for a given light intensity were calculated as the sum of the net oxygen evolution rate and the respiration rates following each light phase. The results were obtained from independent measurements of five biological replicates. Statistical analyses were performed using two-tailed t-test assuming unequal variance with FDR correction by comparing treated samples with the pre-heat samples under the same light.

**Transmission Electron Microscopy (TEM)**. Algal samples (10 mL, around 2×$10^6$ cells/mL) harvested from PBRs were concentrated (1000 g, 2 min, room temperature), drawn into dialysis tubing (Spectrapor®, Spectrum Laboratories, Inc., Cat No. 132294), immersed in 20% (w/v) Bovine Serum Albumin (BSA. Sigma, Cat No. A7030-100G) and immediately frozen in a high-pressure freezer (Leica EM ICE; Leica Microsystems, IL, USA). Freeze substitution was carried out in 2% (v/v) osmium tetroxide in 97%/5% acetone/water (v/v) as follows: −80°C for 3 days, −20°C for 1 day, 4°C for 2 h, and then room temp for 2 h. Samples were washed 4 times in acetone, incubated in 1% thiocarbohydrazide (TCH) (EM Sciences, 21900) in acetone for 1 h, washed 4 times in acetone, incubated 1 h in 2% (v/v) osmium tetroxide and washed 4 times in acetone. Samples were then laced in saturated uranyl acetate in 100% acetone overnight, washed in acetone, transferred to 1:1 ethanol:acetone and stained with saturated lead acetate in 1:1 ethanol: acetone for 2 h. Subsequently, cells were then dehydrated in 100% graded acetone and embedded in hard formulation Epon-Araldite (Embed 812, EM Sciences, Cat No. 14121). Embedments were cut into small blocks and mounted in the vise-chuck of a Reichert Ultracut UCT ultramicrotome (Leica, Buffalo Grove, IL, USA). Ultrathin sections (~60 to 70 nm) were cut using a diamond knife (type ultra 35°C, Diatome), mounted on copper grids (FCFT300-CU-50, VWR, Radnor, PA, USA), and counterstained with lead citrate for 8 min[133]. Sample were imaged with a LEO 912 AB Energy Filter Transmission Electron Microscope (Zeiss, Oberkochen,

Germany). Micrographs were acquired with iTEM software (ver. 5.2) (Olympus Soft Imaging Solutions GmbH, Germany) with a TRS 2048 x 2048k slow-scan charge-coupled device (CCD) camera (TRÖNDLE Restlichtverstärkersysteme, Germany). Thirty electron micrographs were quantified for each time point and treatment. Each TEM image was acquired at 8,000X magnification and 1.37 nm pixel resolution. TEM images were analyzed with Stereology Analyzer software version 4.3.3 to quantify the relative volume mitochondria. Grid type was set as "point" with a sampling step of 500×500 pixels and pattern size of 15×15 pixels. The percent of relative volume for mitochondria was collected after identifying all grid points within one cell and further analyzed in excel[19]. TEM images with a magnification of 8 K were used in the Fiji (ImageJ, FIJI software, National Institutes of Health) analysis. The images were scaled to 0.7299 pixel/nm in ImageJ before the analysis of the pyrenoid area. Two different statistical tests were used to find the significance of p-values. The Kolmogorov-Smirnov test was used for relative volume data since it is commonly used to find significance between data in a form of ratios. A two-tailed t-test with unequal variance was used for area data from ImageJ. All statistical tests compared the treatment conditions to the pre-heat condition.

**Starch quantification**. Starch was extracted and quantified according to modified method from kit (Megazyme, K-TSTA-100A). Frozen sample pellets were homogenized using Qiagen TissueLyser and washed twice in 80% (v/v) ethanol at 85°C. The insoluble fraction was resuspended in DMSO and heated at 110°C for 10 min to improve starch solubilization. Starch hydrolysis and quantification were performed following kit protocol. Starch content was either normalized to cell number or cell volume. Each condition has three biological replicates.

**Statistics and Reproducibility**. All measurements had at least 3 biological replicates. Statistical analyses were conducted as mentioned in each method above. For more than 17 comparisons, FDR using Benjamin-Hochberg correction method was performed with adjusted p value < 0.05 as significance. Source data and p values were included in Supplementary Data 11.

**Reporting summary**. Further information on research design is available in the Nature Research Reporting Summary linked to this article.

## Data availability

The datasets analyzed in this paper are included in this published article and supplementary information files. The RNA-seq data discussed in this publication have been deposited in the NCBI Gene Expression Omnibus[134] and are accessible through GEO accession number GSE182207. The mass spectrometric proteomic data is available via the ProteomeXchange Consortium partner repository, PRIDE[135,136] with the dataset identifier PXD027778. Other information is available from the corresponding author on request.

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

## Acknowledgements

This research was supported by the start-up funding from the Donald Danforth Plant Science Center (DDPSC), the Department of Energy (DOE) Basic Energy Sciences (BES) Photosynthetic Systems (PS) grant (Award No. 0019464), the DOE Biological & Environmental Research (BER) grant (Award No. 0020400), and the RNA-seq service support from the Department of Energy (DOE) Joint Genome Institute (JGI) Community Science Program (CSP, 503414) to R.Z.. The work conducted by the U.S. Department of Energy Joint Genome Institute is supported by the Office of Science of the U.S. Department of Energy under Contract No DE-AC02-05CH11231. E.M.M. was supported by the William H. Danforth Fellowship in Plant Sciences, DDPSC start-up funding (to R.Z.) and Washington University in St. Louis. M.S. was funded by Deutsche Forschungsgemeinschaft (TRR175, projects C02 and D02). We would like to thank researchers from the Kramer Laboratory (Drs. David Kramer, Jeffrey Cruz, Robert Zegarac, Ben Lucker, Geoffry Davis) for their helpful suggestions and discussion to set up and optimize the IDEAspec for spectroscopic measurements in algal cells. Dr. Geoffry Davis is also thanked for his suggestions for 77 K chlorophyll fluorescence measurement. We would like to thank Drs. Jeremy Schmutz, Susannah Tringe, Christa Pennacchio, Chris Daum, and Ronan O'Malley for RNA-seq service at DOE JGI. We acknowledge imaging support from the Advanced Bioimaging Laboratory (RRID:SCR_018951) at DDPSC and the usage of the LEO 912AB Energy Filter TEM acquired through a National Science Foundation (NSF) Major Research Instrumentation grant (DBI-0116650). We also acknowledge proteomics support from the Proteomics and Mass Spectrometry at DDPSC and usage of the Orbitrap Fusion Lumos LC-MS/MS, which was funded by the National Science Foundation under Grant No. DBI-1827534. We would like to thank Dr. James Umen for helpful discussion about our research and Sarah Rommelfanger for

suggestions on RNA-seq analysis. We also want to thank Drs. Xin Wang and Blake Meyers for valuable feedback of the manuscript.

## Author contributions

R.Z. supervised the whole project. R.Z., N.Z. and E.M.M. designed and planned all the experiments. R.Z. and M.S. discussed and designed the time points for high temperature treatments. N.Z. led the project, characterized cell physiologies (including cell density, size, viability, protein, ROS, starch, light microscopic images), conducted RT-qPCRs and extracted RNA for RNA-seq analysis. W.M. and M.X. grew and maintained algal cultures in photobioreactors. N.Z, W.M., C.A. J.J, M.X. and E.M.M. harvested algal samples from photobioreactors with different treatments at different time points. E.M.M. led RNA-seq data analysis and generated all related figures. J.C.B. provided suggestions for RNA-seq analysis and statistical analysis. N.Z. extracted protein for proteomics. S.T. and B.E. quantified protein abundance using LC-MS/MS. B.V., D. Z., T.M., E.M.M., M.S., and N.Z. analyzed the proteomics data. C.C. and J.C. provided suggestions for transcriptomic and proteomic analysis. K.P., F.C and N.Z. performed DNA content analysis. C.A. and M.X. quantified chlorophyll and carotenoid contents. W.M. performed all spectroscopic measurements of photosynthetic parameters. J.J. and L.P. performed $O_2$ evolution measurements. N.Z. and J.J. optimized the ROS protocol. N.Z., E.B. and K.J.C. performed TEM analysis. R.Z., E.M.M., and N.Z. led the writing of the manuscript with the contribution of all other co-authors. R.Z., E.M.M., N.Z., M.S., B.V., M. X., T.M., J.J, K.J.C., J.C.B., K.P, B.E. revised the manuscript. All the authors discussed the results and contributed to data interpretation.

## Competing interests

The authors declare no competing interests.
