## [Peer Review File · Communications Biology]

Reviewers' comments:

Reviewer #1 (Remarks to the Author):

Zhang et al. present the analysis of their large-scale analysis of the green algal *Chlamydomonas reinhardtii* in response to heat stress and recovery from that stress. This is an important topic given expected changing frequency in extreme weather events. There is a lot of data in this manuscript for understanding temporal responses at multi scales in this model alga, and appears to be high-quality with biological replication. Importantly, the multi-omics data has been deposited, and processed data is available in supplemental tables so that this work can serve as a resource for the community (especially non-experts who do not feel comfortable reprocessing the raw data).

Major comments:

(1) These first comments are not asking for more experiments/measurements but are requesting that terms are defined and used accurately: There needs to be clear and explicit definitions of the terms "doubling time" and "generation rate" and "growth rate", especially because the authors have determined cell size differences, cell cycle arrest differences, and chlorophyll content differences.

a. "Doubling time" in this manuscript is defined as the time it takes for the OD680 to double? Since chlorophyll a has a maximum absorbance at 680 nm (as noted by the authors), OD680 is then being used as a proxy for chlorophyll content? So, really, what is being measured is doubling of chlorophyll a per time. This is an important distinction because most microbiologists may assume the authors are suggesting that cell number is doubling.

b. Similar confusion arises when the authors refer to "growth rate". This is usually defined as number of cells per time. But, here, the authors are defining "growth rate" as amount of chlorophyll per time?

c. For Fig 1A, how was number of generations per hour calculated? Since a mother cell can produce 2^n daughters per generation. Was n calculated, constant, and considered in the calculation of "generations per hour"? The authors state, "Doubling time is inverse of relative growth rates and smaller doubling time represents faster growth." True, but the authors are measuring OD680, there is no data showing that for these heat treatments that OD680 is an accurate proxy for "growth", which I and other readers would take to mean number of cells per time. As such, the authors cannot state that in the PBRs that "Algal growth increased" or "decreased", because based on the data presented, the authors can only say that the rate of chlorophyll a production in the PBR increased or decreased.

d. For instance, the authors write: "Algal growth increased during 35C, decreased during 40C, and remained steady at constant 25C (Figure 1B). Increased growth under 35C was confirmed by the medium consumption rates and growth on plates (Supplemental Figure 2)". For clarity this should be written to something as follows: "The rate of chlorophyll a production increased during cultivation at 35C and decreased at 40C compared to 25C. The perceived difference between 35C and 25C was likely due to an increase in cell number per time based on medium consumption rates and growth on plates at these temperatures (Supplemental Figure 2)."

(2) There are a lot of DEGs; a figure showing how many transcripts are annotated by MapMan would clarify how many transcripts are included in the functional interpretation of the data and how many are excluded because of lack of function predictions by MapMan.

(3) In Figure 3, it would be useful to indicate how many genes/proteins are included in the given statistically significant functional terms. Background shading showing the extent of deviation from consensus for each module is needed.

(4) Since the 40C cells contained 4x more chlorophyll per cell and the OD680 was kept constant in the PBR, was the cell density in these PBRs lower (i.e. 4x lower) during the time course?

(5) How many DEGs are specific to green algae or shared with land plants (i.e., no clear ortholog or homolog in land plants broadly and *Arabidopsis* specifically)? This is a very useful and potentially very insightful analysis, especially given the emphasis by the authors on using this data to understand crops or improving thermotolerance of crops (with novel algal genes, for instance).

Minor comments:

(6) The authors state that LHCA1 and TRXF2 "did not change". Although difficult to see because

the y-axis is too large for the plotted data, there do appear to be changes, and no indication if these are statistically significant or not.

(7) Please clarify if the $t=0$ time point is the same as "RHT"? It would help the reader to be consistent and use one or the other in the figures. Same with the use of $T=0$ or "RCT" in different panels and figures.

Reviewer #2 (Remarks to the Author):

Zhang et al provide a comprehensive overview of the reaction of green alga *Chlamydomonas reinhardtii* to types of heat stress, a moderate one at 35°C and an acute one at 40°C. They nicely point down the differences and commonalities between the two types of stress and suggest possible ways how the data can be used to improve algal and possibly plant tolerance to high temperature. The MS is easy to read and follow and it is of possible interest to wide range of readers from algal basic research and biotechnology to plant heat stress response researchers. I generally like the MS and have only several minor comments.

I. 736-740 "The increased gluconeogenesis ..." This sentence implies that the cells at 35 °C were starved and not growing. Yet, within the data there seems to be no proof of the cells being further stressed at 35 °C. The difference in cell size between the 35 °C and 40 °C treatments can be rather attributed to the cell cycle synchronization and cell division at 35 °C. This way small new born cells were formed, which naturally shifted the cell size distribution in the population. In contrast, cell division was arrested but growth was not at 40 °C so the cell size increased. It would be interesting to compare differences in cell numbers between the treatments. The cell numbers at 35 °C should increase significantly once the population has started dividing by 16 h of the treatment judging both from the FACS and cell size data. In contrast, the changes in cell numbers at 40 °C probably will not be as high.

Figure 4 Do the authors know DNA ploidy in recovery samples in between 8 and 24 hours? Alternatively, information on cell numbers in the two data points would be sufficient. Based on the FACS and cell size data it seems that the culture divided between 8 and 24 recovery time points.

Figure 5A At least two cells should be shown for each microscopy image.

I. 1015 Please translate the rpm to g to fit with the rest of the MS.

Reviewer #3 (Remarks to the Author):

Communications Biology 2 Nov 2021
Comments of referee

Author(s): N. Zhang, E.M. Mattoon, W. McHargue, B. Venn, D. Zimmer, K. Pecani, J. Jeong, C.M. Anderson, C. Chen, J.C. Berry, M. Xia, S.-C. Tzeng, E. Becker, L. Pazouki, B. Evans, F. Cross, J. Cheng, K.J. Czymmek, M. Schroda, T. Muhlhaus, R. Zhang

Title: "Systems-wide analysis revealed shared and unique responses to moderate and acute high temperatures in the green alga *Chlamydomonas reinhardtii*"

Manuscript ID: COMMSBIO-21-2716

The manuscript "Systems-wide analysis revealed shared and unique responses to moderate and acute high temperatures in the green alga *Chlamydomonas reinhardtii*" (COMMSBIO-21-2716) presented by N. Zhang and co-authors is a comprehensive system-wide analysis of the differential (moderate versus acute) temperature responses of *Chlamydomonas reinhardtii*. I am impressed by the amount of experimental data included and the attempt to present all findings in a clear and understandable way. The manuscript is well structured and written, but requires some adjustments and some points should be addressed (see comments) before becoming acceptable

for publication in the journal of Communications Biology.

Comments:

1. Some of the major findings/conclusions included in the "Abstract", such as "Heat at 35oC transiently arrested the cell cycle.....and increased starch accumulation." and "Heat at 40oC arrested the cell cycle, inhibited growth, resulting in carbon uptake over usage and increased starch accumulation." (p. 1, lines 23-28) sound quite trivial and in fact similar, if not identical to experimental results reported earlier (see for example: Zachleder V. (2019) *Cells*, 8, 1237; Ivanov I.N. et al. (2021) *Cells*, 10, 1084). I would advise the authors to re-write the "Abstract" and focus on the new findings.

2. Moreover, the background information provided in the "Introduction" section of the manuscript looks quite sketchy and although some of the previous research on heat induced effects on cell cycle is briefly referred to, earlier manuscripts related to heat induced effects on starch accumulation in green algae are not even mentioned. This is not acceptable for me and I would recommend the authors to amend the "Introduction", including all previous research related to their study.

3. What was the measuring temperature for all functional (Chlorophyll fluorescence, electrochromic shift, P700 oxidation, oxygen evolution) measurements for pre-heated, heat treated and recovering from heat-treatment samples? None of the subsections in the "Materials and methods", related to these measurements indicates the measuring temperature for each of the samples. One might assume that all measurements were performed at room temperature and if that is the case, there is a problem. Heat treated samples should have been measured at the corresponding temperatures of 35oC or 40oC. If not, the experimental results for heat-treated samples are highly questionable and the experiments should be performed again at 35oC or 40oC. Please, clarify and correct if necessary.

4. p.3, lines 70-73: "Additionally, high temperature at 40oC can cause damage to photosynthetic electron transport chains, reducing photosynthetic efficiency (Zhang and Sharkey, 2009; Sharkey and Zhang, 2010), and leading to increased reactive oxygen species (ROS) accumulation (Janni et al., 2020)." - High temperatures effects on the photosynthetic electron transport and ROS accumulation have been reported (and reviewed) for decades and way before the references cited in this paragraph. Please, remove them and include more appropriate references.

5. p.3, lines 76-77: "In contrast to the extensive research on the effects during heat, how photosynthetic cells recover from heat is much less studied." - How much "less studied"? Please, include the appropriate references.

6. p.36, line 1092: Low temperature (77K) Chlorophyll fluorescence measurements are very poor, if any, indicator for antenna sizes of both photosystems. PSI/PSII fluorescence ratio measured at 77K rather correlates with photosystems stoichiometry as suggested by the cited reference (Murakami, 1997). In addition, changes in PSI/PSII ratio could be discussed in terms of excitation energy distribution/re-distribution between the two photosystems. The equation used by the authors (p.36, line 1105: $PSII\% = PSII\ peak / (PSII\ peak + PSI\ peak)$) is meaningless and I would advise the authors to remove this section and the associated data from the manuscript.

7. p.37, line 1113: What was the chlorophyll concentration used for oxygen evolution measurements? Please, include this information in the revised version of the manuscript.

We thank the reviewers for reading our manuscript and providing valuable comments. We revised the manuscript according to the suggestions from the reviewers. Please see our responses to each of the comments in purple below. We also tracked the modifications in the text of the revised manuscript.

Reviewers' comments:

Reviewer #1 (Remarks to the Author):

Zhang et al. present the analysis of their large-scale analysis of the green algal *Chlamydomonas reinhardtii* in response to heat stress and recovery from that stress. This is an important topic given expected changing frequency in extreme weather events. There is a lot of data in this manuscript for understanding temporal responses at multi scales in this model alga, and appears to be high-quality with biological replication. Importantly, the multi-omics data has been deposited, and processed data is available in supplemental tables so that this work can serve as a resource for the community (especially non-experts who do not feel comfortable reprocessing the raw data).

We thank the reviewer for the encouraging comment.

Major comments:

(1) These first comments are not asking for more experiments/measurements but are requesting that terms are defined and used accurately: There needs to be clear and explicit definitions of the terms “doubling time” and “generation rate” and “growth rate”, especially because the authors have determined cell size differences, cell cycle arrest differences, and chlorophyll content differences.

a. “Doubling time” in this manuscript is defined as the time it takes for the OD₆₈₀ to double? Since chlorophyll a has a maximum absorbance at 680 nm (as noted by the authors), OD₆₈₀ is then being used as a proxy for chlorophyll content? So, really, what is being measured is doubling of chlorophyll a per time. This is an important distinction because most microbiologists may assume the authors are suggesting that cell number is doubling.

We thank the reviewer for the great suggestion. Yes, OD₆₈₀ is proportional to total chlorophyll content in units of µg chlorophyll per mL. Our algal cultivation method used turbidostatic mode. Fresh medium was added to the culture automatically by a peristaltic pump when the OD₆₈₀ reached the defined maximum value to dilute the culture and the pump was stopped when the OD₆₈₀ dropped to the defined minimum value (Supplemental Figure 1A, B). Algal cultures then grew up at approximately exponential rate to the defined maximum OD₆₈₀ value before the next dilution cycle. The turbidostatic mode precisely controlled the growth condition. We calculated the doubling time or relative growth rates based on the exponential growth phase. The reviewer is right that the doubling time and growth rates we referred to are based on the increase of OD₆₈₀ or the total chlorophyll per mL culture. We modified our figures (main Figure 1A, B and Supplemental Figure 1C) and the first paragraph of the Result section to clarify this.

b. Similar confusion arises when the authors refer to “growth rate”. This is usually defined

as number of cells per time. But, here, the authors are defining “growth rate” as amount of chlorophyll per time?

Yes, we defined growth rates based on the increased OD₆₈₀ or chlorophyll content per mL. We modified our figures and text to clarify this.

c. For Fig 1A, how was number of generations per hour calculated? Since a mother cell can produce 2ⁿ daughters per generation. Was n calculated, constant, and considered in the calculation of “generations per hour”? The authors state, “Doubling time is inverse of relative growth rates and smaller doubling time represents faster growth.” True, but the authors are measuring OD₆₈₀, there is no data showing that for these heat treatments that OD₆₈₀ is an accurate proxy for “growth”, which I and other readers would take to mean number of cells per time. As such, the authors cannot state that in the PBRs that “Algal growth increased” or “decreased”, because based on the data presented, the authors can only say that the rate of chlorophyll a production in the PBR increased or decreased.

We agree with the reviewer. We modified our figures and text to clarify this.

d. For instance, the authors write: “Algal growth increased during 35C, decreased during 40C, and remained steady at constant 25C (Figure 1B). Increased growth under 35C was confirmed by the medium consumption rates and growth on plates (Supplemental Figure 2)”. For clarity this should be written to something as follows: “The rate of chlorophyll a production increased during cultivation at 35C and decreased at 40C compared to 25C. The perceived difference between 35C and 25C was likely due to an increase in cell number per time based on medium consumption rates and growth on plates at these temperatures (Supplemental Figure 2).”

We modified the text based on the reviewer’s suggestion.

(2) There are a lot of DEGs; a figure showing how many transcripts are annotated by MapMan would clarify how many transcripts are included in the functional interpretation of the data and how many are excluded because of lack of function predictions by MapMan.

We thank the reviewer for this great suggestion. We added Supplemental Figures 4G and 4H, showing the number of transcripts and proteins that have MapMan annotations and the number of transcripts and proteins that do not have MapMan annotations. We also added pie charts to the transcriptional and protein network modules in Figure 3 showing the proportion of genes in each module that have MapMan annotations to those that do not have MapMan annotations.

(3) In Figure 3, it would be useful to indicate how many genes/proteins are included in the given statistically significant functional terms. Background shading showing the extent of deviation from consensus for each module is needed.

We thank the reviewer for these great suggestions. We added a ratio in parentheses after each MapMan term shown in Figure 3, showing the number of genes/proteins in that module with the given MapMan term versus the number of genes/proteins in the entire dataset with the given MapMan term. We also added background shading behind each

z-transformed consensus expression pattern showing the standard deviation of z-transformed expression patterns of module members.

(4) Since the 40C cells contained 4x more chlorophyll per cell and the OD₆₈₀ was kept constant in the PBR, was the cell density in these PBRs lower (i.e. 4x lower) during the time course?

Yes, the reviewer is correct. We used turbidostatic mode for algal growth in photobioreactors by monitoring OD₆₈₀, which is proportional to total chlorophyll content per mL. During the experiments, the chlorophyll per mL was maintained near constantly to ensure consistent light illumination and frequent nutrient supply. Cells treated with 40°C had 4x more chlorophyll per cell during heat. Because the constant chlorophyll per mL was maintained during the experiment, algal cultures during 40°C had reduced cell density. We added two figures about chlorophyll per mL and cell density to Supplemental Figure 11K and L. The increased chlorophyll per cell in 40°C treated cells were mainly due to cell division arrest and the rate of chlorophyll increase in 40°C treated cells were slower than that under 35°C or 25°C, thus, showing reduced relative growth rates based on the increase of OD₆₈₀ or chlorophyll increase per mL culture.

(5) How many DEGs are specific to green algae or shared with land plants (i.e., no clear ortholog or homolog in land plants broadly and Arabidopsis specifically)? This is a very useful and potentially very insightful analysis, especially given the emphasis by the authors on using this data to understand crops or improving thermotolerance of crops (with novel algal genes, for instance).

We thank the reviewer for this great suggestion. We used the JGI InParanoid ortholog list to investigate the conservation of Chlamydomonas heat induced genes (HIGs) with *Volvox carteri* (Volvox), *Arabidopsis thaliana* (Arabidopsis), *Oryza sativa* (rice), *Triticum aestivum* (wheat), *Glycine max* (soybean), *Zea mays* (maize), *Sorghum bicolor* (sorghum), and *Setaria viridis* (Setaria). We added Supplemental Figure 5G to show these data. We found that 509 of the Chlamydomonas HIGs have a one-to-one orthologous relationship with Arabidopsis. We added a subsheet to Supplementary Dataset 1 to list these data.

Minor comments:

(6) The authors state that LHCA1 and TRXF2 “did not change”. Although difficult to see because the y-axis is too large for the plotted data, there do appear to be changes, and no indication if these are statistically significant or not.

We mentioned in the Supplemental Figure 1G/H and their figure legends that “Statistical analyses were performed with two-tailed t-test assuming unequal variance by comparing different time points with the first time point. No significance (ns) among different time points ($p > 0.05$).” We also added brackets to Supplemental Figure 1G/H to make this clear.

(7) Please clarify if the t=0 time point is the same as “RHT”? It would help the reader to be consistent and use one or the other in the figures. Same with the use of T=0 or “RCT” in different panels and figures.

We thank the reviewer for the great suggestion. T=0 during heat is when the photobioreactors reached the targeted high temperatures. It took about 20 or 25 min to

reach the high temperature of 35 or 40°C from 25°C (Supplemental Figure 1D), and roughly the same amount of time to cool the photobioreactors from high temperatures to the control temperature of 25°C. The time point when the temperature reached the targeted high temperature of 35 or 40°C was called “RHT” (reach high temperature). Similarly, T=24h during heat is the end of heating and just before we switched photobioreactors from high temperatures to the control temperature at 25°C. The time point when the temperature decreased to the control temperature of 25°C was called “RCT” (reach control temperature 25°C for recovery), equivalent to T=0 during the recovery. We modified the manuscript and modified related figures (Main Figure 2E/F, 3, 4A, Table 1, Supplemental Figure 4A/B, 5, 6, 9, 10) to reduce confusion. We replaced the labels of RHT (reach high temperature) with T=0 during heat and replaced the labels of RCT (reach control temperature 25°C for recovery) with T=0 during recovery. The time points during heat and recovery were marked with different colors for easy understanding, brown/red colors for time points during heat and blue colors for time points during recovery. We also modified related figure legends to reflect these changes.

Reviewer #2 (Remarks to the Author):

Zhang et al provide a comprehensive overview of the reaction of green alga *Chlamydomonas reinhardtii* to types of heat stress, a moderate one at 35°C and an acute one at 40°C. They nicely point down the differences and commonalties between the two types of stress and suggest possible ways how the data can be used to improve algal and possibly plant tolerance to high temperature. The MS is easy to read and follow and it is of possible interest to wide range of readers from algal basic research and biotechnology to plant heat stress response researchers. I generally like the MS and have only several minor comments.

We thank the reviewer for the encouraging comment.

I. 736-740 “The increased gluconeogenesis ...” This sentence implies that the cells at 35 °C were starved and not growing. Yet, within the data there seems to be no proof of the cells being further stressed at 35 °C.

The sentence the reviewer referred to is “The increased gluconeogenesis and glyoxylate cycles may contribute to the increased growth under 35°C heat; however, the faster-growing cell population with increased carbon metabolism probably experience starvation under nutrient limiting conditions thus reducing biomass accumulation and cell growth in relatively short time.”

We used turbidostatic mode for all algal cultivation in liquid in this manuscript. Fresh medium was added to the culture automatically by a peristaltic pump when the OD₆₈₀ reached the defined maximum value to dilute the culture and the pump was stopped when the OD₆₈₀ dropped to the defined minimum value (Supplemental Figure 1A, B). Our OD₆₈₀ range was sufficiently small that we expected the minimal nutrient limitation during our experiment. Under our experimental conditions with sufficient nutrient supply, we think the increased growth at 35°C could partially be explained by the increased gluconeogenesis and glyoxylate cycles for increased acetate uptake. The sentence the

reviewer mentioned was to speculate what would happen in the field condition where nutrients are limited. Under nutrient limiting conditions, the increased carbon metabolisms by moderate heat of 35°C may result in cell starvation, and, eventually, reduced growth and biomass accumulation. We have another manuscript in preparation which confirmed this hypothesis. We modified the sentence above to prevent confusion, shown below, and moved this modified section to the part of discussion regarding starch accumulation, gluconeogenesis, and glyoxylate cycle.

“Furthermore, we used turbidostatic mode for algal cultivation in liquid with frequent influx of fresh medium supply so the nutrient limitation during our experiments would be minimal. Thus, the higher acetate uptake through the increased gluconeogenesis and glyoxylate cycles may contribute to the faster growth under 35°C heat. However, in nature, nutrients, especially environmental carbon sources, are often limited. We speculate that the cell populations with increased carbon metabolisms under moderate high temperatures probably experience starvation under nutrient limiting conditions in the field, thus eventually reducing biomass accumulation and cell growth under prolonged moderate heat like 35°C.”

The difference in cell size between the 35 °C and 40 °C treatments can be rather attributed to the cell cycle synchronization and cell division at 35 °C. This way small new born cells were formed, which naturally shifted the cell size distribution in the population. In contrast, cell division was arrested but growth was not at 40 °C so the cell size increased.

The reviewer is correct. Cell cycle was transiently inhibited during early heat of 35°C but later around 8-h of heat at 35°C, cells were able to overcome the cell cycle inhibition and resume the cell cycle (Figure 4A, 5A-C). The moderate high temperature at 35°C represents a unique condition where we could observe the dynamic of cell cycle inhibition and subsequent recovery during the heat treatment.

It would be interesting to compare differences in cell numbers between the treatments. The cell numbers at 35 °C should increase significantly once the population has started dividing by 16 h of the treatment judging both from the FACS and cell size data. In contrast, the changes in cell numbers at 40 °C probably will not be as high.

We thank the reviewer for the suggestion. We added the cell density data to Supplemental Figure 11L. We used turbidostatic mode for algal growth in photobioreactors by monitoring OD₆₈₀, which is proportional to total chlorophyll content per mL. During the experiments, the chlorophyll per mL was maintained near constantly to ensure consistent light illumination and frequent nutrient supply. Cells treated with 40°C had 4x more chlorophyll per cell during heat. Because the constant chlorophyll per mL was maintained during the experiment, algal cultures during 40°C had reduced cell density. Cells treated with 35°C had transiently increased chlorophyll, thus transiently decreased cell density. However, the doubling of chlorophyll was much faster at 35°C than 25 or 40°C, showing increased relative growth rates based on the rate of increased OD₆₈₀. Due to the turbidostatic control of our photobioreactors, we could not measure cell division based on cell density in the culture. However, our data in Figures 4 and 5 supported the division of cells around 16-h of 35°C heat treatment, as the reviewer suggested.

Figure 4 Do the authors know DNA ploidy in recovery samples in between 8 and 24 hours? Alternatively, information on cell numbers in the two data points would be sufficient. Based on the FACS and cell size data it seems that the culture divided between 8 and 24 recovery time points.

The reviewer is right. Based on the DNA content data and cell size data, cells treated with 40°C started to divide around 8-h of recovery. The DNA ploidy information in recovery samples between 8 and 24 hours after 40°C heat treatment can be seen in Figure 4Q, S, U. Figure 4A (expression of cell cycle related transcripts) and Figure 5 (A-C, cell images and cell size data) also support this conclusion.

Figure 5A At least two cells should be shown for each microscopy image.

We modified Figure 5A to show more cells for each microscopy image. The cell density we used in these experiments were around or below 2×10^6 cells mL⁻¹ and our microscopy images were from high magnification (63x (NA1.4) oil-immersion objective with a Leica DMI6000 B microscope). For the purposes of displaying all time points together in one panel, we could not use very large imaging areas, thus some images only had one cell. The microscopy images shown are representative cell images, and detailed quantification can be seen in Figure 5B-D.

I. 1015 Please translate the rpm to g to fit with the rest of the MS.

Great suggestion! We changed rpm to g in the method session about pigment analysis.

Reviewer #3 (Remarks to the Author):

Communications Biology 2 Nov 2021
Comments of referee

Author(s): N. Zhang, E.M. Mattoon, W. McHargue, B. Venn, D. Zimmer, K. Pecani, J. Jeong, C.M. Anderson, C. Chen, J.C. Berry, M. Xia, S.-C. Tzeng, E. Becker, L. Pazouki, B. Evans, F. Cross, J. Cheng, K.J. Czymmek, M. Schroda, T. Muhlhaus, R. Zhang

Title: "Systems-wide analysis revealed shared and unique responses to moderate and acute high temperatures in the green alga *Chlamydomonas reinhardtii*"

Manuscript ID: COMMSBIO-21-2716

The manuscript "Systems-wide analysis revealed shared and unique responses to moderate and acute high temperatures in the green alga *Chlamydomonas reinhardtii*" (COMMSBIO-21-2716) presented by N. Zhang and co-authors is a comprehensive system-wide analysis of the differential (moderate versus acute) temperature responses of *Chlamydomonas reinhardtii*. I am impressed by the amount of experimental data included and the attempt to present all findings in a clear and understandable way.

We thank the reviewer for the encouraging comment.

The manuscript is well structured and written, but requires some adjustments and some points should be addressed (see comments) before becoming acceptable for publication in the journal of Communications Biology.

Comments:

1. Some of the major findings/conclusions included in the "Abstract", such as "Heat at 35°C transiently arrested the cell cycle.....and increased starch accumulation." and "Heat at 40°C arrested the cell cycle, inhibited growth, resulting in carbon uptake over usage and increased starch accumulation." (p. 1, lines 23-28) sound quite trivial and in fact similar, if not identical to experimental results reported earlier (see for example: Zachleder V. (2019) *Cells*, 8, 1237; Ivanov I.N. et al. (2021) *Cells*, 10, 1084). I would advise the authors to re-write the "Abstract" and focus on the new findings.

Zachleder et al. 2019 and Ivanov et al. 2021 showed the inhibition of cell cycle and increased of starch in *Chlamydomonas reinhardtii* with 39°C treatment. However, our experimental conditions were quite different from these two papers. (1) We grew cultures under mixotrophic condition with supplied carbon source, and air level 0.04% CO₂; but they used autotrophic condition without supplied carbon source with 2% CO₂. (2) We used 100 μmol photons m⁻² s⁻¹ light (50% red: 50% blue, provided by light-emitting diode panel), but Zachleder et al. 2019 used 500 μmol photons m⁻² s⁻¹ light (provided by light-emitting diode lamps) and Ivanov et al. 2021 used 50 μmol photons m⁻² s⁻¹ light (provided by luminescent lamps) in addition to natural sunlight. (3) We used unsynchronized cultures, but they used synchronized cultures; (4) We used 25°C as the control temperature, but they used 30°C as the control temperature; (5) We used turbidostatic mode with frequent nutrient supply and well controlled conditions, but they did not have fresh medium supply during the experiment. Our results about cell cycle inhibition and starch accumulation under high temperatures were consistent with these two papers despite different experimental conditions, suggesting these phenomena are independent of carbon source, CO₂ levels, light intensities, synchrony of the cultures, and the control temperature. These two papers and our work are complementary to each other, and all contribute to understanding algal heat responses. Additionally, we provided system-wide analysis to try to elucidate the mechanisms of these phenomena. Furthermore, the transient inhibition of cell cycle under moderate heat of 35°C reported in our work was not covered in Zachleder et al. 2019 or Ivanov et al. 2021 because they only used acute high temperature of 39°C, which completely inhibited the cell cycle. Combining our analysis at multiple levels, we concluded that the increased gluconeogenesis/glyoxylate-cycles most likely contributed to the increased growth and starch accumulation under 35°C but the inhibited cell division may explain the reduced growth but increased starch accumulation under 40°C. Nevertheless, we thank the reviewer for the suggestion, and we modified the abstract to remove the inhibited cell cycle and starch accumulation under 40°C.

2. Moreover, the background information provided in the "Introduction" section of the manuscript looks quite sketchy and although some of the previous research on heat induced effects on cell cycle is briefly referred to, earlier manuscripts related to heat induced effects on starch accumulation in green algae are not even mentioned. This is

not acceptable for me and I would recommend the authors to amend the "Introduction", including all previous research related to their study.

We thank the reviewer for the suggestion. The reviewer is right that we should cite the two papers, Zachleder et al. 2019 and Ivanov et al. 2021, which mentioned the heat induced cell cycle arrested and starch accumulation. We modified the introduction to add these two references. Due to page limitation, we were not able to cover all previous research related to algal heat responses, but only included highly relevant literatures to our research.

3. What was the measuring temperature for all functional (Chlorophyll fluorescence, electrochromic shift, P700 oxidation, oxygen evolution) measurements for pre-heated, heat treated and recovering from heat-treatment samples? None of the subsections in the "Materials and methods", related to these measurements indicates the measuring temperature for each of the samples. One might assume that all measurements were performed at room temperature and if that is the case, there is a problem. Heat treated samples should have been measured at the corresponding temperatures of 35oC or 40oC. If not, the experimental results for heat-treated samples are highly questionable and the experiments should be performed again at 35oC or 40oC. Please, clarify and correct if necessary.

All the photosynthetic parameters were measured at the room temperature at 25°C for algal cultures sampled at different time points with different temperature treatments. We thank the reviewer for this suggestion and added the information to the method section.

Regarding the reviewer's view about "measured at the corresponding temperatures of 35°C or 40°C", we have some different thoughts. We sampled algal cultures at different time points with different temperature treatments and performed photosynthetic measurements at 25°C to investigate how the changes induced by high temperatures affected algal photosynthetic performance by comparing with algal cultures treated and measured at 25°C. If we perform the photosynthetic measurements at corresponding high temperatures, it will complicate the results. (1) The photosynthetic measurements performed at high temperatures cannot be compared directly with those measured at 25°C, unless the cultures used for measurements at high temperatures and 25°C have identical treatments. At high temperatures of 35°C or 40°C, the mechanical parameters of the photosynthetic measurement devices may change. For example, both O₂ concentration in liquid phase and the oxygen electrode disc are temperature sensitive. Therefore, all photosynthetic measurements must be collected at the same temperature to be directly compared with each other. To do so, we would be required to measure photosynthetic parameters at 35°C or 40°C in cultures that have grown under the control condition of 25°C, which is not reflective of the true control photosynthetic parameters. Instead, our measurements of high temperature treated cultures at 25°C reflect the heat-induced changes to photosynthetic parameters by comparing with those from cultures grown and also measured at 25°C. (2) We currently do not have the ability to measure photosynthetic parameters inside photobioreactors with heating. We need separate devices to monitor photosynthetic parameters, which have limitations for under which temperatures they can operate.

Most of the previous high temperature experiments performed photosynthetic measurements in heat-treated samples at room temperatures. Please see the references

below. We appreciate the thoughts from the reviewer, but based on the reasoning above, we believe our photosynthetic measurements at room temperature are valid.

Hemme et al., 2014, *The Plant Cell*, Systems-Wide Analysis of Acclimation Responses to Long-Term Heat Stress and Recovery in the Photosynthetic Model Organism *Chlamydomonas reinhardtii*. (Chlorophyll fluorescence and O₂ evolution from heat treated cells were measured at 25°C).

Tanaka et al. 2000, *Plant Physiology*. Acclimation of the Photosynthetic Machinery to High Temperature in *Chlamydomonas reinhardtii* Requires Synthesis de Novo of Proteins Encoded by the Nuclear and Chloroplast Genomes. (O₂ evolution from heat treated cells were measured at 25°C).

Anderson et al. 2021, *Communications biology*. High Light and Temperature Reduce Photosynthetic Efficiency through Different Mechanisms in the C₄ model *Setaria viridis*. (Chlorophyll fluorescence and gas exchange in heat treated leaves were measured at 25°C).

4. p.3, lines 70-73: "Additionally, high temperature at 40oC can cause damage to photosynthetic electron transport chains, reducing photosynthetic efficiency (Zhang and Sharkey, 2009; Sharkey and Zhang, 2010), and leading to increased reactive oxygen species (ROS) accumulation (Janni et al., 2020)." - High temperatures effects on the photosynthetic electron transport and ROS accumulation have been reported (and reviewed) for decades and way before the references cited in this paragraph. Please, remove them and include more appropriate references.

We appreciate the suggestion from the reviewer and added more references and revised the statement. The references we cited in the introduction cannot cover all previous high temperature research. The references we cited supported the heat effects at 40°C to which we specifically referred, and we did not mean to cover all different high temperatures that were conducted in the literature.

5. p.3, lines 76-77: "In contrast to the extensive research on the effects during heat, how photosynthetic cells recover from heat is much less studied." - How much "less studied"? Please, include the appropriate references.

We thank the reviewer for the great suggestion. We added the reference and modified the statement.

6. p.36, line 1092: Low temperature (77K) Chlorophyll fluorescence measurements are very poor, if any, indicator for antenna sizes of both photosystems. PSI/PSII fluorescence ratio measured at 77K rather correlates with photosystems stoichiometry as suggested by the cited reference (Murakami, 1997). In addition, changes in PSI/PSII ratio could be discussed in terms of excitation energy distribution/re-distribution between the two photosystems. The equation used by the authors (p.36, line 1105: $PSII\% = PSII\ peak / (PSII\ peak + PSI\ peak)$) is meaningless and I would advise the authors to remove this section and the associated data from the manuscript.

Our 77K chlorophyll fluorescence data is comparable to that published previously in *Chlamydomonas reinhardtii*. Please see these references below.

Hemme et al., 2014, *The Plant Cell*, Systems-Wide Analysis of Acclimation Responses to Long-Term Heat Stress and Recovery in the Photosynthetic Model Organism *Chlamydomonas reinhardtii* (Figure 3).

RØKKE et al. 2018. *Photosynthetica*. Chlorophyll Fluorescence Emission Spectroscopy of Oxygenic Organisms at 77 K (Figure 9C).

Our 77K measurements followed the similar protocol as in Luker and Kramer, 2013 and produced highly reproducible results (Supplemental Figure 14F). We also validated our measurements by using State 1 and State 2 conditions as Luker and Kramer, 2013 and got similar results.

Luker and Kramer, 2013, *Photosynthesis Research*. Regulation of Cyclic Electron Flow in *Chlamydomonas reinhardtii* under Fluctuating Carbon Availability.

We agree with the reviewer that changes in PSI/PSII ratio could be due to light redistribution between the two photosystems. Our formula, $PSII\% = \frac{PSII \text{ peak}}{PSII \text{ peak} + PSI \text{ peak}}$, used normalized PSII and PSI peaks (Spectral data were normalized to the PSII spectral maximum value at 686 nm, as mentioned in the method and as in published literatures). The calculated PSII% estimates the fraction of light that is used by PSII. We used this ratio to calculate the linear electron flow rate, $LEF = (\Phi_{PSII}) \cdot (PSII\%) \cdot (\text{fraction of absorbed light})$. We appreciate the suggestion from the reviewer, but believe our data is valid and would like to keep these data if possible because they showed how high temperatures affected the light distribution between PSII and PSI.

7. p.37, line 1113: What was the chlorophyll concentration used for oxygen evolution measurements? Please, include this information in the revised version of the manuscript. We thank the reviewer for the great suggestion. We used 2 mL of cells (around 10 μg chlorophyll) for oxygen evolution measurements. We added the information in the revised method section.

REVIEWERS' COMMENTS:

Reviewer #1 (Remarks to the Author):

no additional comments

Reviewer #2 (Remarks to the Author):

The revised version of manuscript of Zhang et al. contains some new information as well as some clarifications of several statements. Unfortunately, some of the new statements over-interpret the data such as in:

- 1) The statement replacing the discussed sentence on increased gluconeogenesis and glyoxylate (Furthermore, we used turbidostatic mode for algal cultivation in liquid with frequent influx of fresh medium supply so the nutrient limitation during our experiments would be minimal. Thus, the higher acetate uptake through the increased gluconeogenesis and glyoxylate cycles may contribute to the faster growth under 35°C heat. However, in nature, nutrients, especially environmental carbon sources, are often limited. We speculate that the cell populations with increased carbon metabolisms under moderate high temperatures probably experience starvation under nutrient limiting conditions in the field, thus eventually reducing biomass accumulation and cell growth under prolonged moderate heat like 35°C.). Firstly, the data provide no proof of higher acetate uptake at 35°C. In the contrary, supplementary figure 9 G shows no change in acetate uptake. This might be solved by changing the wording to perhaps: Thus, the increased gluconeogenesis and glyoxylate cycles may contribute to the faster growth under 35°C heat. The other part of the statement starting from "we speculate" is indeed speculation that cannot be derived from the data. In principle, it is impossible to directly compare relatively dense culture grown at defined and stable growth conditions (irrespective of the heat treatment) to natural conditions. No such extrapolation is possible. It has been established that combination of stresses will have different outcome than each of the stresses alone. This indeed might be the case for the combination of moderate heat at 35°C and nutrient starvation mentioned by the authors in rebuttal letter. But this does not justify the speculation. Also I would advise on moving the statement to different place of discussion or better connecting it to the rest of the text as now its meaning is rather clouded.
- 2) In several places of the MS (e. g. l. 200-202, and later in discussion) there it is now referred to increase in cell number per time. Yet, no proof of this is shown in the MS. The supplementary figure 2 showing plate assay which was evaluated after 44 hours and 3 days proves that growth improves at 35°C but it has no direct relation to the behavior of the culture during the treatment at 35°C. In the treated culture, only chlorophyll content was measured which is not a proof of cell division since the division could be (partially or temporarily) inhibited without affecting the growth. Moreover, this even seems to be the case in the experiment in question as it is claimed elsewhere in the MS. Due to the methodological approach chosen the authors are unable to directly refer to changes in cell number. This is perfectly fine as this has been already established elsewhere but the unnecessary over-statements should be avoided. Indeed, the growth improvement at higher temperature is not novel in *Chlamydomonas reinhardtii*. It was noted by Lien and Knutsen (1979) and was thoroughly analyzed for wide range of temperatures (Vítová et al., 2011), including the 25 and 35°C in question in the MS. The growth improvement with temperature has been established in terms of growth rates, cell size increase and cell number changes. Furthermore, there are other places where the data are over/mis-interpreted.
- 3) In several places in discussion, it is mentioned that ROS production was increased at 40°C, which is used as basis for discussion of the correlation between ROS levels and DNA replication and ROS levels and the increased level of chlorophyll. It is true that ROS levels were higher at 40°C heat treatment if normalized to cell number. Yet, due to the extensive changes in cell volume during the 40°C treatment, the values cannot be compared at face values. Instead, they should be normalized to cell volume as presented in Supplementary figure 11. The values normalized per cell number presented in Figure 5 should be replaced by their alternatives from Supplementary figure 11 as the volume normalized comparison is more biologically relevant. Correspondingly, the statements referring to the ROS levels should be re-phrased to reflect the lack of difference in ROS levels between the treatments.
- 4) In discussion, it is implied that the recoveries between the 35 and 40°C heat treatments are different since they have differently regulated genes for photosynthesis light reaction. Here again,

the two types of cultures cannot be compared at face value. Clearly, the cultures are different not only in the terms of cell size as mentioned above but also physiological state. The culture recovering from 35°C heat was partially synchronized by the treatment, the cells (partially synchronously) divided during the treatment and the newly born cells have been growing since then, i.e. for at least 8 hours already during the 35°C treatment. The 40°C heat treated culture was instead heat arrested, grew to large size but was unable to replicate DNA and divide as shown by the authors on Figures 4 and 5. It only became able to re-start the cell cycle after moving to recovery temperature. This can explain why it is behaving similarly to the synchronized culture at the early dark phase when it starts to divide as it is physiologically very close to that culture. There is no reason to believe that the cell cycle completion (DNA replication and cell division) (and the connected changes in gene expression, etc.) during the recovery of this heat synchronized culture is different from the day/night synchronized culture at the same physiological time. Yet, it is to be expected it will be different from the 35°C heat treated culture that is in different physiological time, at least 8 hours off (but likely more) of the 40°C heat culture.

Minor points:

- 5) At several occasion (e. g. l. 789) it is mentioned that growth was inhibited by the 40°C treatment. Yet, this is not supported by the data. The growth rate decreased, the growth was slower and compromised but it was not totally inhibited as is clear from both the chlorophyll and cell size data so the wording should reflect this to avoid misunderstanding.
- 6) l. 801-803 The statement is misleading. It is not clear if this entire sentence refers both to the MS data and to Zachleder et al., 2019 or to the published paper alone. Should the reference be only to the Zachleder et al. paper it should be amended to claim the increased chlorophyll to 18-h or 29-h as it is stated in the paper instead of 8-h heat.

- 1) Lien, T., and Knutsen, G. (1979). Synchronous growth of *Chlamydomonas reinhardtii* (Chlorophyceae): a review of optimal conditions. *Journal of Phycology* 15, 191-200.
- 2) Vítová, M., Bišová, K., Hlavová, M., Kawano, S., Zachleder, V., and Čížková, M. (2011). *Chlamydomonas reinhardtii*: duration of its cell cycle and phases at growth rates affected by temperature. *Planta* 234, 599-608.

We thank the reviewers for reading our revised manuscript and providing valuable comments. We revised the manuscript according to the suggestions from the Reviewer #2. Please see our responses to each of the comments in purple below. We also tracked the modifications in the text of the revised manuscript.

Reviewers' comments:

Reviewer #2 (Remarks to the Author):

The revised version of manuscript of Zhang et al. contains some new information as well as some clarifications of several statements. Unfortunately, some of the new statements over-interpret the data such as in:

1) The statement replacing the discussed sentence on increased gluconeogenesis and glyoxylate (Furthermore, we used turbidostatic mode for algal cultivation in liquid with frequent influx of fresh medium supply so the nutrient limitation during our experiments would be minimal. Thus, the higher acetate uptake through the increased gluconeogenesis and glyoxylate cycles may contribute to the faster growth under 35°C heat. However, in nature, nutrients, especially environmental carbon sources, are often limited. We speculate that the cell populations with increased carbon metabolisms under moderate high temperatures probably experience starvation under nutrient limiting conditions in the field, thus eventually reducing biomass accumulation and cell growth under prolonged moderate heat like 35°C.).

Firstly, the data provide no proof of higher acetate uptake at 35°C. In the contrary, supplementary figure 9 G shows no change in acetate uptake. This might be solved by changing the wording to perhaps: Thus, the increased gluconeogenesis and glyoxylate cycles may contribute to the faster growth under 35°C heat.

We thank the reviewer for this great suggestion. Most transcripts related to acetate uptake/assimilation did not change significantly during 35°C heat, but several proteins related to acetate uptake/assimilation were increased significantly for multiple time points during 35°C heat (Supplementary_dataset_2_proteomics_overview, 35C_DEx sub-sheet and the summary table below). We added new figures to show the dynamic of transcripts/proteins related to acetate uptake/assimilation to Supplementary Figure 8 e-h. In *Chlamydomonas*, acetate uptake feeds into the glyoxylate cycle and gluconeogenesis for starch biosynthesis (Johnson and Alric, 2012, 2013). Our data showed that the increase of proteins related to acetate uptake/assimilation was consistent with the increased proteins related to gluconeogenesis and glyoxylate during 35°C heat (Supplementary Figure 8 a-d), providing evidence for the increased acetate uptake/assimilation, gluconeogenesis and the glyoxylate cycle during 35°C heat. Meanwhile, we followed the reviewer's suggestions to revise the discussion related to acetate uptake/assimilation in this manuscript. We also removed the heatmap of transcripts related to acetate uptake/assimilation in Supplementary Figure 9G (last version) to prevent redundancy because the new figures were added to Supplementary Figure 8.

Function group	Gene	Gene name	Proteins during 35°C heat
Acetate uptake	Cre17.g700750	GFY3	significantly up at RHT, 35°C_4h, 16, 24h
Acetate assimilation	Cre01.g055408	ACSS	significantly up at 35°C_4h, 8h, 16, 24h
Acetate assimilation	Cre01.g071662	ACS1	significantly up at 35°C_4h, 8h, 16, 24h
Acetate assimilation	Cre07.g353450	ACS3	significantly up at 35°C_4h, 8h, 16, 24h
Acetate assimilation	Cre17.g709850	ACK2	significantly up at 35°C_4h, 8h, 16, 24h

The other part of the statement starting from “we speculate” is indeed speculation that cannot be derived from the data. In principle, it is impossible to directly compare relatively dense culture grown at defined and stable growth conditions (irrespective of the heat treatment) to natural conditions. No such extrapolation is possible. It has been established that combination of stresses will have different outcome than each of the stresses alone. This indeed might be the case for the combination of moderate heat at 35°C and nutrient starvation mentioned by the authors in rebuttal letter. But this does not justify the speculation. Also I would advise on moving the statement to different place of discussion or better connecting it to the rest of the text as now its meaning is rather clouded.

We thank the reviewer for this great suggestion. We revised the manuscript as the reviewer suggested.

2) In several places of the MS (e. g. l. 200-202, and later in discussion) there it is now referred to increase in cell number per time. Yet, no proof of this is shown in the MS. The supplementary figure 2 showing plate assay which was evaluated after 44 hours and 3 days proves that growth improves at 35°C but it has no direct relation to the behavior of the culture during the treatment at 35°C. In the treated culture, only chlorophyll content was measured which is not a proof of cell division since the division could be (partially or temporally) inhibited without affecting the growth. Moreover, this even seems to be the case in the experiment in question as it is claimed elsewhere in the MS. Due to the methodological approach chosen the authors are unable to directly refer to changes in cell number. This is perfectly fine as this has been already established elsewhere but the unnecessary over-statements should be avoided.

We thank the reviewer for this great suggestion. The sentence the reviewer referred to (e. g. l. 200-202 in the last submission) is “The perceived difference between 35°C and 25°C was likely due to an increase in cell number per time based on medium consumption rates and growth on plates at these temperatures (Supplemental Figure 2)”. This was actually a suggestion from Reviewer #1. Please check the first round of review comments of Reviewer #1. Because the transient and moderate increase of chlorophyll per cell and cell volume during 35°C heat in addition to the small OD₆₈₀ range we used for turbidostatic control of algal cultivation in photobioreactors, OD₆₈₀ was proportional to both chlorophyll content and cell density during 35°C heat. However, considering Reviewer #2’s

suggestion, we revised this sentence to “The increased growth at 35°C was confirmed by medium consumption rates and growth on plates (Supplementary Figure 2).”

Indeed, the growth improvement at higher temperature is not novel in *Chlamydomonas reinhardtii*. It was noted by Lien and Knutsen (1979) and was thoroughly analyzed for wide range of temperatures (Vítová et al., 2011), including the 25 and 35°C in question in the MS. The growth improvement with temperature has been established in terms of growth rates, cell size increase and cell number changes.

1) Lien, T., and Knutsen, G. (1979). Synchronous growth of *Chlamydomonas reinhardtii* (Chlorophyceae): a review of optimal conditions. *Journal of Phycology* 15, 191-200.
2) Vítová, M., Bišová, K., Hlavová, M., Kawano, S., Zachleder, V., and Čížková, M. (2011). *Chlamydomonas reinhardtii*: duration of its cell cycle and phases at growth rates affected by temperature. *Planta* 234, 599-608.

We thank the reviewer for mentioning these two papers! The Lien and Knutsen (1979) paper does not focus on algal heat responses, but the optimal condition to grow synchronized algal cultures. In Figure 1b of this 1979 paper, they showed relative growth rate as function of temperature in synchronized algal cultures, which is consistent with our data, although we used different experimental conditions. Vítová et al., 2011 reported how different temperatures and day/night cycles affected cell number, cell size, growth rates, and cell cycle length in synchronized algal cultures under photoautotrophic conditions. Our experimental conditions were quite different from this paper. We used non-synchronized algal cultures grown in mixotrophic conditions with turbidostatic controls to maintain cell density and reduce nutrient limitation. Vítová et al., 2011 and our work are complementary to each other, and both contribute to the understanding of algal heat responses. While Vítová et al., 2011 focused on cell division and cell cycle, we thoroughly investigated the temperature effects at multiple levels, including photosynthesis, cellular structures, transcriptomes, proteomes in addition to the cell physiologies mentioned in the Vítová et al., 2011 paper (e.g., cell number, cell size, growth rates). Combining our data at multiple levels, we tried to elucidate the mechanisms of increased growth at 35°C and reduced growth at 40°C. We cited these two references in our revised manuscript.

Furthermore, there are other places where the data are over/mis-interpreted.
3) In several places in discussion, it is mentioned that ROS production was increased at 40°C, which is used as basis for discussion of the correlation between ROS levels and DNA replication and ROS levels and the increased level of chlorophyll. It is true that ROS levels were higher at 40°C heat treatment if normalized to cell number. Yet, due to the extensive changes in cell volume during the 40°C treatment, the values cannot be compared at face values. Instead, they should be normalized to cell volume as presented in Supplementary figure 11. The values normalized per cell number presented in Figure 5 should be replaced by their alternatives from Supplementary figure 11 as the volume normalized comparison is more biologically relevant. Correspondingly, the statements referring to the ROS levels should be re-phrased to reflect the lack of difference in ROS levels between the treatments.

We thank the reviewer for the suggestion. The measured ROS levels reflect the competition between ROS production and scavenging. Our data showed the induction of ROS scavenging transcripts during and after 40°C treatment (Supplementary Figure 9e), which suggested the increased ROS production in 40°C-treated cells. The up-regulated ROS scavengers during and after 40°C treatment most likely were functional to eliminate ROS. Consequently, we did not see much difference in measured ROS levels, but it still pointed to the increased ROS production at the time when the scavenging enzymes were up-regulated in 40°C-treated cells. The reviewer suggested “The values normalized per cell number presented in Figure 5 should be replaced by their alternatives from Supplementary figure 11”. Other parameters in Figure 5 are presented as per cell quantification so we think the ROS per cell volume quantification does not fit Figure 5. The ROS per cell volume data is included in Supplementary Figure 11 and readers can see it easily if they are interested. We discussed this with the editor and got her consensus on this arrangement.

4) In discussion, it is implied that the recoveries between the 35 and 40°C heat treatments are different since they have differently regulated genes for photosynthesis light reaction. Here again, the two types of cultures cannot be compared at face value. Clearly, the cultures are different not only in the terms of cell size as mentioned above but also physiological state. The culture recovering from 35°C heat was partially synchronized by the treatment, the cells (partially synchronously) divided during the treatment and the newly born cells have been growing since then, i.e. for at least 8 hours already during the 35°C treatment. The 40°C heat treated culture was instead heat arrested, grew to large size but was unable to replicate DNA and divide as shown by the authors on Figures 4 and 5. It only became able to re-start the cell cycle after moving to recovery temperature. This can explain why it is behaving similarly to the synchronized culture at the early dark phase when it starts to divide as it is physiologically very close to that culture. There is no reason to believe that the cell cycle completion (DNA replication and cell division) (and the connected changes in gene expression, etc.) during the recovery of this heat synchronized culture is different from the day/night synchronized culture at the same physiological time. Yet, it is to be expected it will be different from the 35°C heat treated culture that is in different physiological time, at least 8 hours off (but likely more) of the 40°C heat culture.

We thank the reviewer for the great discussion. In synchronized algal cultures under day/night cycles, genes related to DNA synthesis and cell cycle peak during the early dark phase when the genes related to photosynthetic light reactions had minimal expression (Zones et al., 2015; Strenkert et al., 2019). We refer the differentially regulated transcripts related to cell cycle and photosynthetic light reactions during the dark phase as uncoupling of these two processes. Because we used algal cultures grown under constant light conditions, we think transcripts related to these two processes may need to express simultaneously. We agree with the reviewer that 35°C heat can synchronize algal cultures, as evidenced by Figure 4. After 35°C heat treatment, the induction of cell

cycle genes was comparable to that after 40°C heat treatment but there was no down-regulation of transcripts related to photosynthetic light reactions in 35°C-treated cells, which may be related to the constant light and moderate heat treatment of 35°C we used. In contrast, 40°C inhibited cell cycle and may not synchronize the cultures as 35°C did. After 40°C heat treatment, transcripts related to cell cycle were up-regulated but transcripts related to photosynthetic light reactions were down-regulated, similar uncoupling of the two processes as the dark phase of day/night cycle synchronized cultures. While unsynchronized cells after 40°C heat had increased ROS level, synchronized cells at the dark phase of day/night cycle most likely have minimal ROS production, as evidenced by the down-regulation of many ROS response genes (Zones et al., 2015). We disagree with the reviewer that “This can explain why the 40°C treated cell is behaving similarly to the synchronized culture at the early dark phase when it starts to divide as it is physiologically very close to that culture.” Cells with 40°C heat treatment have very different physiologies from synchronized culture at the early dark phase under the control temperature, considering the heat damaged cellular structures and biological processes. Thus, we think the uncoupling of transcripts related to cell cycle and photosynthetic light reactions are different between the recovery from 40°C and during the dark phase of day/night cycle. We revised the discussion to clarify the confusion.

Minor points:

5) At several occasion (e. g. l. 789) it is mentioned that growth was inhibited by the 40°C treatment. Yet, this is not supported by the data. The growth rate decreased, the growth was slower and compromised but it was not totally inhibited as is clear from both the chlorophyll and cell size data so the wording should reflect this to avoid misunderstanding.

We thank the reviewer for the great suggestion. We revised the statement as “Heat at 35°C stimulated growth but 40°C decreased growth”. The algal growth at 40°C was largely reduced and the whole algal cultures died after 2-day of heat treatment at 40°C.

6) l. 801-803 The statement is misleading. It is not clear if this entire sentence refers both to the MS data and to Zachleder et al., 2019 or to the published paper alone. Should the reference be only to the Zachleder et al. paper it should be amended to claim the increased chlorophyll to 18-h or 29-h as it is stated in the paper instead of 8-h heat.

We thank the reviewer for the great suggestion. The sentence the reviewer referred to is “Chlamydomonas cells treated at 39°C for more than one day had initially increased chlorophyll (8-h heat) followed by chlorophyll loss, cell bleaching, and death (33-h heat) (Zachleder et al., 2019).” This sentence refers to the data of Figure 3 in Zachleder et al., 2019. Their figure showed the photomicrographs of Chlamydomonas cells grown at 30°C and 39°C. At 29 h of 39°C heat, cells started to lose chlorophyll. After 33 h of 39°C heat, the majority of the cells lost chlorophyll. We revised the sentence to “Chlamydomonas cells treated at 39°C for more than one day had initially increased chlorophyll (8~16-h

heat) followed by chlorophyll loss, cell bleaching, and death (33-h heat) (Zachleder et al., 2019)".